# Existence of processes violating causal inequalities on time-delocalised subsystems

Julian Wechs [1,2] ✉, Cyril Branciard [2] ✉ & Ognyan Oreshkov[1] ✉

It has been shown that it is theoretically possible for there to exist quantum and classical processes in which the operations performed by separate parties do not occur in a well-defined causal order. A central question is whether and how such processes can be realised in practice. In order to provide a rigorous framework for the notion that certain such processes have a realisation in standard quantum theory, the concept of time-delocalised quantum subsystem has been introduced. In this paper, we show that realisations on time-delocalised subsystems exist for all unitary extensions of tripartite processes. This class contains processes that violate causal inequalities, i.e., that can generate correlations that witness the incompatibility with definite causal order in a device-independent manner, and whose realisability has been a central open problem. We consider a known example of such a tripartite classical process that has a unitary extension, and study its realisation on time-delocalised subsystems. We then discuss this finding with regard to the assumptions that underlie causal inequalities, and argue that they are indeed a meaningful concept to show the absence of a definite causal order between the variables of interest.

The concept of causality is essential for physics and for our perception of the world in general. Our usual understanding is that events take place in a definite causal order, with past events influencing future events, but not vice versa. One may however wonder whether this notion is really fundamental, or whether scenarios without such an underlying order can exist. In particular, the questions of what quantum theory implies for our understanding of causality, and what new types of causal relations arise in the presence of quantum effects, have recently attracted substantial interest. This investigation is motivated both by foundational and by applied questions. On the one hand, it is expected to lead to new conceptual insights into the tension between general relativity and quantum theory[1–3]. On the other hand, it also opens up new possibilities in quantum information processing[4].

A particular model for the study of quantum causal relations is the process matrix framework[2], where one considers multiple parties which perform operations that locally abide by the laws of quantum theory, but that are not embedded into any a priori causal order. As it turns out, this framework indeed allows for situations where the causal order between the parties is not well-defined (see e.g. refs. [2,5–9]). Moreover, some of these processes, called noncausal, can produce correlations that violate causal inequalities[2,6–8,10,11], which witnesses the incompatibility with definite causal order in a device-independent manner, similarly to the way a violation of a Bell inequalities witnesses the incompatibility with local hidden variables[12]. A central question is which of these processes with indefinite causal order have a practical realisation, and in what physical situations they can occur. It has been speculated that indefinite causal order could arise in exotic physical regimes, such as at the interface of quantum theory and gravity[1–3]. However, there are also processes with indefinite causal order that have an interpretation in terms of standard quantum theoretical concepts. A paradigmatic example is the quantum switch[4], a process in which the order between two operations is controlled coherently by a two-dimensional quantum system. This control qubit may be prepared in a superposition state, which leads to a superposition of causal orders. Although the quantum switch cannot violate causal inequalities[5,6,13,14] (however, see recent results in the presence of

[1]QuIC, Ecole Polytechnique de Bruxelles, C.P. 165, Université Libre de Bruxelles, 1050 Brussels, Belgium. [2]Univ. Grenoble Alpes, CNRS, Grenoble INP, Institut Néel, 38000 Grenoble, France. ✉e-mail: julian.wechs@ulb.be; cyril.branciard@neel.cnrs.fr; ognyan.oreshkov@ulb.be

additional causal assumptions[15,16]), it can be proven incompatible with a definite causal order in a device-dependent sense[5,6].

In order to demonstrate indefinite causal order in practice, a number of experiments that realise such coherent control of orders have been implemented in the laboratory[17–27], however their interpretation as genuine realisations of indefinite causal order has remained controversial[28–31]. Indeed, the claim that indefinite causal order can be realised in standard quantum scenarios seems contradictory at first sight—after all, such experiments admit a description in terms of standard quantum theory, where physical systems by assumption evolve with respect to a fixed background time, and it is therefore not manifest how the causal order between operations could possibly be indefinite. A resolution of this apparent contradiction was proposed in ref. [29], where it was shown that certain processes with indefinite causal order can be seen to take place as part of standard quantum mechanical evolutions if the latter are described in terms of suitable systems. The twist is to consider a more general type of system than usually studied, namely so-called time-delocalised subsystems, which are nontrivial subsystems of composite systems whose constituents are associated with different times. This concept provides a rigorous underpinning for the interpretation of previous laboratory experiments as realisations of processes with indefinite causal order— when the experiment is described with respect to such an alternative, operationally equally meaningful factorisation of the Hilbert space, it acquires precisely the form of the process with indefinite causal order. It was then shown in ref. [29] that this argument extends to an entire class of quantum processes, namely unitary extensions of bipartite processes, as well as a class of isometric extensions, whose relation to the unitary class is not yet fully understood. The generalisation of these constructions to more parties has however remained an open question. In particular, it has remained an open question whether processes violating causal inequalities can be realised in a similar way. It is in fact generally believed that such processes could not be realised deterministically within the known physics[14].

In this paper, we extend the proof of realisability on time-delocalised subsystems to all unitary extensions of tripartite processes. This class contains examples of processes that can violate causal inequalities, showing that they have realisations with the tools of known physics in a well-defined sense.

This work is structured as follows. We set the stage by reviewing the process matrix framework, as well as the notion of time-delocalised subsystems. We present the general tripartite construction, and we study an example of a tripartite noncausal process on time-delocalised subsystems. We then analyse our finding with regard to the assumptions that underlie causal inequalities, and argue that their violation witnesses the absence of a definite causal order in a meaningful way.

## Results

### Notations
We start by introducing some notations. We denote the Hilbert space of some quantum system $Y$ by $\mathcal{H}^Y$, the dimension of $\mathcal{H}^Y$ by $d_Y$ and the space of linear operators over $\mathcal{H}^Y$ by $\mathcal{L}(\mathcal{H}^Y)$. Each such Hilbert space comes with a preferred, computational basis generally denoted $\{|i\rangle^Y\}_i$. The identity operator on $\mathcal{H}^Y$ is denoted by $\mathbb{1}^Y$. We also use the notation $\mathcal{H}^{YZ} := \mathcal{H}^Y \otimes \mathcal{H}^Z$ for the tensor product of two Hilbert spaces $\mathcal{H}^Y$ and $\mathcal{H}^Z$ (whose computational basis is built as the tensor product of the two subsystems' computational bases). For two isomorphic Hilbert spaces $\mathcal{H}^Y$ and $\mathcal{H}^Z$, we denote the identity operator between these spaces (i.e. the canonical isomorphism, which maps each computational basis state $|i\rangle^Y$ of $\mathcal{H}^Y$ to the corresponding computational basis state $|i\rangle^Z$ of $\mathcal{H}^Z$) by $\mathbb{1}^{Y \to Z} := \sum_i |i\rangle^Z \langle i|^Y$, and its pure Choi representation (see the Methods section "The Choi isomorphism and the link product") by $|\mathbb{1}\rangle\rangle^{YZ} := \sum_i |i\rangle^Y \otimes |i\rangle^Z$. (Generally, superscripts on vectors indicate the Hilbert space they belong to, which may be omitted when clear from the context). Moreover, we will often abbreviate $X_I X_O$

to $X_{IO}$ for the incoming and outgoing systems of the party $X$ (see below).

### The process matrix framework
In the following, we briefly outline the process matrix framework, originally introduced in ref. [2]. We consider multiple parties $X = A, B, C, \ldots$ performing operations that are locally described by quantum theory. That is, each party has an incoming quantum system $X_I$ with Hilbert space $\mathcal{H}^{X_I}$ and an outgoing quantum system $X_O$ with Hilbert space $\mathcal{H}^{X_O}$, and can perform arbitrary quantum operations from $X_I$ to $X_O$. A quantum operation is most generally described by a quantum instrument, that is, a collection of completely positive (CP) maps $\{\mathcal{M}_X^{[o_X]}\}_{o_X}$, with each $\mathcal{M}_X^{[o_X]} : \mathcal{L}(\mathcal{H}^{X_I}) \to \mathcal{L}(\mathcal{H}^{X_O})$ associated to a classical outcome $o_X$, and with the sum over the classical outcomes yielding a completely positive and trace-preserving (CPTP) map.

The process matrix framework was conceived to study the most general correlations that can arise between such parties, without making any a priori assumption about the way they are connected. In ref. [2], it was shown that these correlations can most generally be expressed as

$$P(o_A, o_B, o_C, \ldots) = W * \left( M_A^{[o_A]} \otimes M_B^{[o_B]} \otimes M_C^{[o_C]} \otimes \cdots \right). \tag{1}$$

Here, $M_X^{[o_X]} \in \mathcal{L}(\mathcal{H}^{X_{IO}})$ are the Choi representations of the local CP maps $\mathcal{M}_X^{[o_X]}$ and "$*$" denotes the link product[32,33], a mathematical operation that describes the composition of quantum operations in terms of their Choi representation (see the Methods section "The Choi isomorphism and the link product"). $W \in \mathcal{L}(\mathcal{H}^{A_{IO}B_{IO}C_{IO}\cdots})$ is a Hermitian operator called the process matrix. The requirement that the probabilities in Eq. (1) should be non-negative, even when the operations of the parties are extended so as to act on additional, possibly entangled ancillary input systems, is equivalent to $W \geq 0$. The requirement that the probabilities should be normalised (i.e., they should sum up to 1 for any choice of local operations) is equivalent to $W$ satisfying certain linear constraints[2,5,6,9,34], and having the trace $\text{Tr}(W) = d_{A_O} d_{B_O} d_{C_O} \cdots$.

The process matrix is the central object of the formalism, which describes the physical resource or environment through which the parties are connected. Mathematically, the process matrix defines (i.e., it is the Choi representation of) a quantum channel $\mathcal{W} : \mathcal{L}(\mathcal{H}^{A_O B_O C_O \cdots}) \to \mathcal{L}(\mathcal{H}^{A_I B_I C_I \cdots})$ from all output systems of the parties to their input systems. Equation (1) then describes the composition of that channel with the local operations, which can be interpreted as a circuit with a cycle as represented graphically (for the bipartite case) in Fig. 1a.

Through the top-down approach outlined here, one recovers standard quantum scenarios, such as joint measurements on multipartite quantum states, or, more generally, quantum circuits in which the parties apply their operations in a fixed causal order (and the process matrix corresponds to the acyclic circuit fragment consisting of the operations in between the parties[33,35]). However, one also finds processes that are incompatible with any definite causal order between the local operations. Such processes are said to be causally nonseparable[2,5,6,9]. Furthermore, some causally nonseparable processes can generate correlations $P(o_A, o_B, o_C, \ldots | i_A, i_B, i_C, \ldots)$, where $i_X$ are local classical inputs based on which the local operations are chosen, that violate so-called causal inequalities[2,6–8,10,11], which certifies their incompatibility with a definite causal order in a device-independent way. Such processes are referred to as noncausal.

A class of processes that is of particular interest in this paper is that of unitarily extendible processes, which were first discussed in ref. [34]. A unitary extension of a process matrix $W$ is a process matrix which involves an additional party $P$ with a trivial, one-dimensional input Hilbert space, as well as an additional party $F$ with a trivial, one-dimensional outgoing Hilbert space, such that the corresponding channel from $P_O A_O B_O C_O \ldots$ to $F_I A_I B_I C_I \ldots$ is unitary, and such that the original process matrix $W$ is recovered when $P$ prepares some fixed

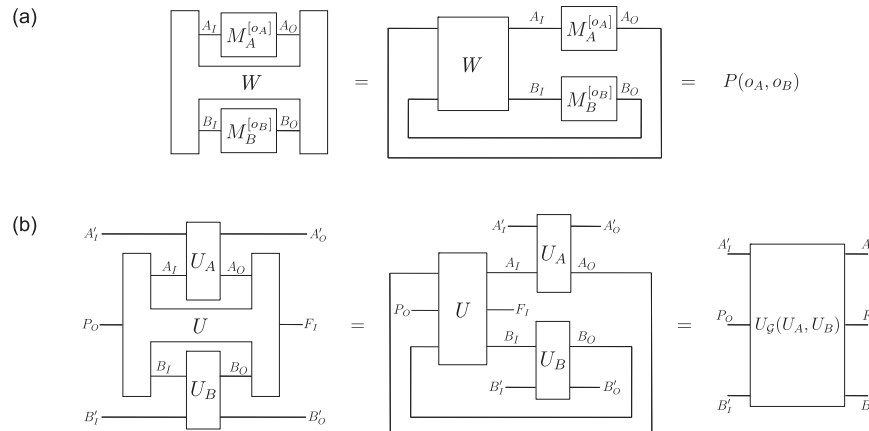

**Fig. 1 | Process matrix scenarios as cyclic circuits. a** In the process matrix framework, the operations performed by the parties (here, Alice and Bob) are composed with the process matrix, which defines a channel from the output systems $A_O B_O$ of the parties back to their input systems $A_I B_I$. This composition can be seen as a cyclic circuit, and provides the probabilities for the classical outcomes $o_A$ and $o_B$. **b** Composing a unitarily extended process matrix with unitary operations performed by the parties gives rise to a unitary operation from the outgoing system $P_O$ of the global past party $P$ and the incoming ancillas of the parties $A'_I, B'_I$ to the incoming system $F_I$ of the global future party $F$ and the outgoing ancillas of the parties $A'_O, B'_O$[34].

state and $F$ is traced out. That is, the extended process matrix is a rank-one projector $|U\rangle\rangle\langle\langle U|$, where $|U\rangle\rangle$ is the pure Choi representation (see Methods) of a unitary $U : \mathcal{H}^{P_O A_O B_O C_O \cdots} \rightarrow \mathcal{H}^{F_I A_I B_I C_I \cdots}$, which satisfies

$$W = |U\rangle\rangle\langle\langle U| * (|0\rangle\langle 0|^{P_O} \otimes \mathbb{1}^{F_I}). \tag{2}$$

The additional parties $P$ and $F$ can be interpreted as being in the global past, respectively global future, of all other parties, since they do not receive, respectively send out, a quantum system.

Note that the unitary extension also needs to be a valid process matrix, i.e., it needs to satisfy the above-mentioned constraints which ensure that it yields valid probabilities when the parties (including $P$ and $F$) perform arbitrary local operations. In ref. [34], it was found that some process matrices do not admit such a unitary extension, and unitary extendibility was postulated as a necessary condition for a process matrix to describe a physically realisable scenario. It was also shown that unitary extensions are equivalent to processes that preserve the reversibility of quantum operations. That is, when the slots of $P$ and $F$ are left open, and all other parties perform unitary operations $\mathcal{U}_X : \mathcal{L}(\mathcal{H}^{X_I X'_I}) \rightarrow \mathcal{L}(\mathcal{H}^{X_O X'_O})$, which act on $X_I$ and $X_O$ as well as some (possibly trivial) additional ancillary incoming and outgoing systems $X'_I$ and $X'_O$, the resulting global operation, which takes the initial systems $P_O A'_I C'_I \ldots$ to the final systems $F_I A'_O B'_O C'_O \ldots$, is again unitary (Fig. 1b).

In this case, the Choi representations of the local operations, as well as the unitarily extended process matrix, are rank-one projectors, and we can describe their composition in terms of their pure Choi representations. The global unitary operation $\mathcal{U}_\mathcal{G}(\mathcal{U}_A, \mathcal{U}_B, \mathcal{U}_C, \cdots) : \mathcal{L}(\mathcal{H}^{P_O A'_I B'_I C'_I \cdots}) \rightarrow \mathcal{L}(\mathcal{H}^{F_I A'_O B'_O C'_O \cdots})$, in its pure Choi representation, is given by

$$\left| U_\mathcal{G}(U_A, U_B, U_C, \cdots) \right\rangle\rangle = |U\rangle\rangle * \left( |U_A\rangle\rangle \otimes |U_B\rangle\rangle \otimes |U_C\rangle\rangle \otimes \ldots \right) \\ \in \mathcal{H}^{P_O A'_I B'_I C'_I \cdots F_I A'_O B'_O C'_O \cdots} \tag{3}$$

where $|U_X\rangle\rangle \in \mathcal{H}^{X_{IO} X'_{IO}}$ are the pure Choi representations of the local unitary operations $\mathcal{U}_X$, and "$*$" denotes here the so-called vector link product[13] (cf. Methods). In the following, we are going to refer to $|U\rangle\rangle$ as the process vector of the unitary process under consideration.

The process matrices that we are concerned with in this work are unitary extensions of bipartite or tripartite process matrices. Moreover, any local operation can be dilated to a unitary channel acting on the original incoming and outgoing systems together with an additional incoming and outgoing ancilla, followed by a measurement of

the outgoing ancilla. Throughout the paper, we will therefore not consider the actions of the global past and global future parties explicitly, but rather work with the description as per Eq. (3) in terms of pure Choi representations, which is convenient. We will also take the incoming and outgoing Hilbert spaces of all parties, except for $P$ and $F$, to be of equal dimension $d_{X_I} = d_{X_O} =: d$. This simplification saves us some technicalities, and it does not entail any loss of generality. Namely, if these dimensions do not match, one can treat the process under consideration as a part of an extended process with process vector $|U\rangle\rangle \otimes |\mathbb{1}\rangle\rangle^{P_A \bar{A}_I} \otimes |\mathbb{1}\rangle\rangle^{\bar{A}_O F_A} \otimes |\mathbb{1}\rangle\rangle^{P_B \bar{B}_I} \otimes |\mathbb{1}\rangle\rangle^{\bar{B}_O F_B} \otimes |\mathbb{1}\rangle\rangle^{P_C \bar{C}_I} \otimes |\mathbb{1}\rangle\rangle^{\bar{C}_O F_C} \otimes \ldots$, which involves additional identity channels between additional outgoing (incoming) Hilbert spaces $\mathcal{H}^{P_A}, \mathcal{H}^{P_B}, \mathcal{H}^{P_C}, \ldots$ ($\mathcal{H}^{F_A}, \mathcal{H}^{F_B}, \mathcal{H}^{F_C}, \ldots$) of the global past (future) party, and additional incoming (outgoing) Hilbert spaces $\mathcal{H}^{\bar{A}_I}, \mathcal{H}^{\bar{B}_I}, \mathcal{H}^{\bar{C}_I}, \ldots$ ($\mathcal{H}^{\bar{A}_O}, \mathcal{H}^{\bar{B}_O}, \mathcal{H}^{\bar{C}_O}, \ldots$) of the remaining parties, whose dimensions are chosen such that $d_{X_I \bar{X}_I} = d_{X_O \bar{X}_O} = d$ for all parties (except $P$ and $F$).

## Time-delocalised subsystems and operations

In this section, we discuss the concept of time-delocalised subsystem, first introduced in ref. [29]. Briefly summarised, the idea is that a quantum circuit, consisting of operations that act at definite times on specific input and output systems, can be described in terms of a different choice of systems, corresponding to an alternative factorisation of the joint Hilbert spaces of the input and output systems of operations at different times. In general, the new systems may be delocalised relative to the old ones and thus spread over different times. When described in terms of such alternative time-delocalised subsystems, the circuit generally contains cycles as considered in the process matrix framework (Fig. 1). We first discuss the general formalisation of this idea, then we recall how it applies to the case of the quantum switch, as well as general unitary extensions of bipartite processes, for which it was shown in ref. [29] that realisations on such time-delocalised subsystems always exist.

The concept of time-delocalised subsystem arises from combining two notions from standard quantum theory, namely the definition of quantum subsystem decompositions in terms of tensor product structures, and the fact that a fragment of a quantum circuit containing multiple operations implements itself a quantum operation from all its incoming to all its outgoing systems.

In quantum theory, the division of a physical system into subsystems is formally described through the choice of a tensor product structure. Equipping a given Hilbert space $\mathcal{H}^Y$, corresponding to some quantum system $Y$, with a tensor product structure means choosing an

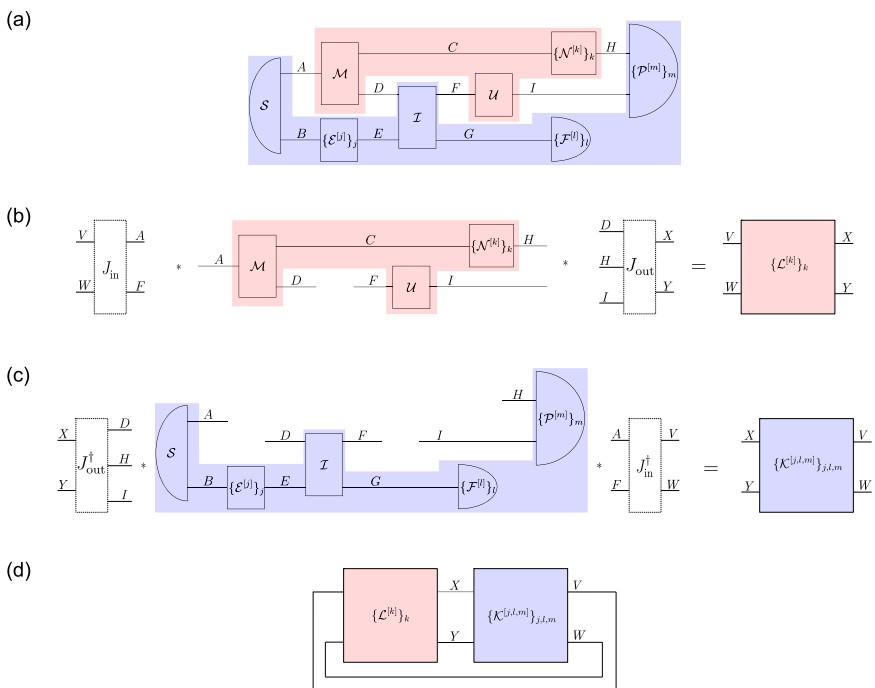

**Fig. 2 | Description of a quantum circuit in terms of time-delocalised subsystems. a** Example of a quantum circuit, consisting of quantum operations $\mathcal{S}, \mathcal{M}, \{\mathcal{E}^{[j]}\}_j, \mathcal{I}, \mathcal{U}, \{\mathcal{N}^{[k]}\}_k, \{\mathcal{F}^{[l]}\}_l, \{\mathcal{P}^{[m]}\}_m$, which are composed through the systems $A, B, C, D, E, F, G, H, I$, and a decomposition thereof into fragments, corresponding to the red and blue boxes. The red fragment implements itself a quantum operation from the incoming systems $A$ and $F$ to the outgoing systems $D, H$ and $I$, which are each associated with different times. It is composed with its complement, the blue fragment, which implements a quantum operation from the systems $D, H, I$ to the systems $A, F$. **b** Description of the red circuit fragment in terms of time-delocalised subsystems $V, W, X, Y$, which are defined by isomorphisms $J_{\text{in}} : \mathcal{H}^{VW} \to \mathcal{H}^{AF}$ and $J_{\text{out}} : \mathcal{H}^{DHI} \to \mathcal{H}^{XY}$. We obtain a new operation $\{\mathcal{L}^{[k]}\}_k$ from $V, W$ to $X, Y$. **c** Description of the blue circuit fragment in terms of the time-delocalised subsystems $V, W, X, Y$. We obtain a new operation $\{\mathcal{K}^{[j,l,m]}\}_{j,l,m}$ from $X, Y$ to $V, W$. **d** In the new subsystem description in terms of the time-delocalised subsystems $V, W, X, Y$, we obtain a cyclic circuit composed of $\{\mathcal{L}^{[k]}\}_k$ and $\{\mathcal{K}^{[j,l,m]}\}_{j,l,m}$.

isomorphism (i.e., a unitary transformation) $J : \mathcal{H}^Y \to \otimes_{i=1}^n \mathcal{H}^{Y_n}$, where $\mathcal{H}^{Y_1}, \ldots, \mathcal{H}^{Y_n}$ are Hilbert spaces of dimensions $d_{Y_1}, \ldots, d_{Y_n}$, with $\Pi_{i=1}^n d_{Y_n} = d_Y$. Such a choice establishes a notion of locality on $\mathcal{H}^Y$, and defines a decomposition of the system $Y$ into subsystems $Y_1, \ldots, Y_n$. For instance, the operators in $\mathcal{L}(\mathcal{H}^Y)$ that are local on the subsystem $Y_i$ are those of the form $J^\dagger(O^{Y_i} \otimes \mathbb{1}^{Y_1, \ldots, Y_{i-1} Y_{i+1} \ldots Y_n})J$ with $O^{Y_i} \in \mathcal{L}(\mathcal{H}^{Y_i})$. (Equivalently, the tensor product structure can also be defined in terms of the algebras of operators that are local on the different subsystems[36]). Since the choice of such a tensor product structure is not unique, there are many different ways to view $\mathcal{H}^Y$ as the state space of some quantum system with multiple subsystems.

In standard quantum theory, physical systems evolve with respect to a fixed background time. At an abstract level, such standard quantum mechanical time evolution can be described in terms of a quantum circuit, that is, a collection of quantum operations (pictorially represented by boxes) that are composed through quantum systems (pictorially represented by wires) in an acyclic network. The operations in such a quantum circuit thus act on their incoming and outgoing quantum systems (which may consist of several subsystems) at definite times. One may however also consider quantum operations that act on several subsystems associated with different times. In fact, this possibility arises naturally within the quantum circuit framework. Namely, if one considers a generic fragment of a quantum circuit containing many operations, that fragment implements a quantum operation from the joint system of all wires that enter into it, to the joint system of all wires that go out of it[33], where the incoming and outgoing wires are generally associated with Hilbert spaces at different times (see Fig. 2a for an example).

One may choose to describe such a quantum operation implemented by a fragment with respect to a different subsystem decomposition. Formally, this is achieved by composing its incoming, respectively outgoing, wires with some isomorphisms that define a different tensor product structure on the corresponding joint Hilbert spaces (Fig. 2b). The resulting subsystems are then in general not associated with a definite time. This is what one understands by time-delocalised subsystems.

To describe the full circuit in terms of these newly chosen time-delocalised subsystems, the operation implemented by the complement of the fragment under consideration needs to be composed with precisely the inverse of the chosen isomorphisms (Fig. 2c). The composition of the two fragments (which, for a circuit with no open wires, corresponds to the joint probability of the measurement outcomes of the different operations in the circuit[37,38], see Fig. 1a) then indeed remains the same in the old and new descriptions. This follows from the properties of the link product (see Methods, Eqs. (13) and (14)), which provides a formal tool to connect the different fragments that a quantum circuit is decomposed into[32,33].

Importantly, the structure of a given circuit with respect to such a choice of time-delocalised subsystems can also be tested operationally[29]. In particular, the circuit can be disconnected at the chosen subsystems and each of the time-delocalised operations that occur on these subsystems can be experimentally addressed and verified, similarly to the way one would test the circuit description with respect to the standard time-local factorisation. In this sense, such an alternative description of the experiment is operationally just as meaningful. This is discussed in more detail in Supplementary Note 1.

## Processes with indefinite causal order on time-delocalised subsystems

In the laboratory experiments that have been proposed as implementations of the quantum switch, one considers a target quantum system at two possible times. The operation $U_A$ is applied to the target

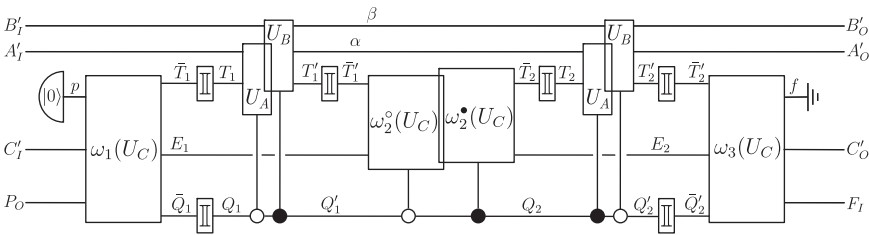

**Fig. 3 | Temporal circuit for a general tripartite unitary process.** $U_A$ and $U_B$ are applied either on the time-local target system $T_1^{(\prime)}$ or $T_2^{(\prime)}$ (and the ancillary systems), depending coherently on the state of the two-dimensional control systems $Q_1^{(\prime)}$ and $Q_2^{(\prime)}$. These two applications of the coherently controlled operations $U_A$ and $U_B$ are surrounded by circuit operations $\omega_1(U_C) : \mathcal{H}^{pC_iP_O} \to \mathcal{H}^{\bar{T}_1E_1\bar{Q}_1}$, $\omega_2^\circ(U_C) : \mathcal{H}^{\bar{T}_1'E_1} \to \mathcal{H}^{\bar{T}_2E_2}$, $\omega_2^\bullet(U_C) : \mathcal{H}^{T_1E_1} \to \mathcal{H}^{\bar{T}_2E_2}$ (these two also being coherently controlled), and $\omega_3(U_C) : \mathcal{H}^{\bar{T}_2'E_2\bar{Q}_2} \to \mathcal{H}^{fC_OF_I}$, which can (together with the therein introduced ancillary systems $E_1, E_2$) in general all depend on $U_C$, the third party's (Charlie's) operation. The boxes $\mathbb{I}$ stand for identity channels that relate the systems with and without the bars. The ancillary system $p$ is prepared in the state $|0\rangle^p$ in the beginning, and the final ancillary system $f$ is discarded. (Note that, with a slight abuse of notation, we use the ground symbol for this discarding of $f$, which is commonly used for mixed circuits where the boxes represent CP maps, rather than for circuits consisting of pure operations, as we have here. The system $f$ however always ends up in the state $|0\rangle^f$ (see Supplementary Note 3A), so that taking the partial trace over $f$ is equivalent to projecting onto $|0\rangle^f$, and does not introduce any decoherence or loss of purity. The coherently controlled applications of $U_A$ and $U_B$, as well as of $\omega_2^\circ(U_C)$ and $\omega_2^\bullet(U_C)$, are displayed with a slight shift for graphical clarity, but they can be taken to act at the same time.

system $T_1$ at the earlier time, or to the target system $T_2$ at the later time, depending on whether another two-dimensional quantum system, the control qubit, is in the computational basis state $|0\rangle$ or $|1\rangle$, and conversely for the operation $U_B$. There has been much debate (see e.g. refs. [28–31]) about whether experiments of that type can be interpreted as valid realisations of the quantum switch, understood as an abstractly defined scenario in the process matrix formalism[5]. Indeed, the relation between the above outlined experimental procedure, and the situation considered in the process matrix framework, where one instance of each $U_A$ and $U_B$ is composed with the process matrix in a circuit with a cycle, is a priori unclear. A heuristic argument that is sometimes invoked to justify that each of the two operations is indeed applied once and only once is that each operation occurs precisely once in each of the two superposed coherent branches, and is therefore used once overall. To further corroborate this, one could introduce a flag or counter system[14,39] that keeps track of the usage of the operations. To really understand the sense in which the quantum switch is realised in these experiments, it is however desirable to rigorously formalise the link between the standard quantum description of the experiments, and the process matrix scenario. This question was addressed in ref. [29]. It was shown that the temporally ordered quantum circuit that describes the experimental situation outlined above indeed takes the structure of a circuit with a cycle as in the process matrix framework (i.e., as in Fig. 1), when one changes to a description in terms of specific time-delocalised subsystems–whose choice, broadly speaking, formalises the intuition that the input system is $T_1$ when the control system is in state $|0\rangle$ and $T_2$ when the control system is in state $|1\rangle$, and similarly for the output systems[29]. In other words, when these experiments are realised physically, what happens on these alternative systems is precisely the structure described in the process matrix framework. It is in that sense that these experiments can be considered realisations of the abstract mathematical concept.

It was then shown that this argument can be generalised, and that there exist other types of processes which have a realisation in this sense. Notably, this is the case for the entire class of unitary extensions of bipartite processes, of which the quantum switch is a particular example. It was subsequently shown in refs. [40,41] that all such processes are variations of the quantum switch, but the proof of Ref. [29] did not rely on this knowledge. It is the idea behind the original proof from ref. [29], together with the subsequent result of refs. [40,41], that will allow us to generalise the proof to the tripartite case. We therefore recall the bipartite result from ref. [29], in the language and conventions we use in this paper (notably employing the Choi representation and the link product), in Methods, and the corresponding proofs in Supplementary Note 2.

## Unitary extensions of tripartite processes on time-delocalised subsystems

For unitary extensions of processes with more than two parties, it is a priori unclear whether and how a realisation on time-delocalised subsystems can be found. In the following, we will establish the result for unitary extensions of tripartite processes. Briefly summarised, we show that for any unitarily extended tripartite process, there exists a standard, temporally ordered quantum circuit, with operations that depend on the local operations $U_A$, $U_B$ and $U_C$ applied in the process, which precisely corresponds to the situation considered in the process matrix framework, with one instance of each $U_A$, $U_B$ and $U_C$ composed with the process matrix in a circuit with a cycle, when described in terms of a suitable choice of time-delocalised subsystems.

Formally, we prove the following proposition.

**Proposition 1.** Consider a unitary extension of a tripartite process, described by a process vector $|U\rangle\rangle \in \mathcal{H}^{P_OA_{IO}B_{IO}C_{IO}F_I}$, composed with unitary local operations $U_A : \mathcal{H}^{A_IA_I'} \to \mathcal{H}^{A_OA_O'}$, $U_B : \mathcal{H}^{B_IB_I'} \to \mathcal{H}^{B_OB_O'}$ and $U_C : \mathcal{H}^{C_IC_I'} \to \mathcal{H}^{C_OC_O'}$. For any such process, the following exist.

1. A temporal circuit of the form shown in Fig. 3, in which $U_A$ and $U_B$ are applied on the target input and output systems $T_1^{(\prime)}$ or $T_2^{(\prime)}$, coherently conditioned on the state of the control systems $Q_1^{(\prime)}$ and $Q_2^{(\prime)}$, and which is composed of circuit operations that depend on $U_C$.
2. Isomorphisms $J_{in} : \mathcal{H}^{A_IB_IC_IYZ} \to \mathcal{H}^{T_1T_2\bar{T}_1'\bar{T}_2'Q_1P_O}$ and $J_{out} : \mathcal{H}^{T_1'T_2'\bar{T}_1\bar{T}_2Q_2'F_I} \to \mathcal{H}^{A_OB_OC_O\bar{Y}\bar{Z}}$, such that, with respect to the subsystems $A_I$, $B_I$ and $C_I$ of $T_1T_2\bar{T}_1'\bar{T}_2'Q_1P_O$ and the subsystems $A_O$, $B_O$ and $C_O$ of $T_1'T_2'\bar{T}_1\bar{T}_2Q_2'F_I$ that these isomorphisms define, the circuit in Fig. 3 takes the form of a cyclic circuit composed of $U, U_A, U_B$ and $U_C$ as in the process matrix framework (Fig. 4).

In the following, we outline the proof. All technical proofs and calculations for this tripartite construction are given in Supplementary Note 3.

**Outline of proof.** The existence of a temporal circuit as in Fig. 3 is shown in Supplementary Note 3A. It follows from the result that all unitary extensions of bipartite processes can be implemented as variations of the quantum switch[40,41], in which the time of the two local operations is controlled coherently. Any unitary extension of a tripartite process can thus be implemented as a variation of the quantum switch, with two local operations whose time is controlled coherently, and which is composed of circuit operations that depend on the third local operation. The isomorphisms $J_{in} : \mathcal{H}^{A_IB_IC_IYZ} \to \mathcal{H}^{T_1T_2\bar{T}_1'\bar{T}_2'Q_1P_O}$ and $J_{out} : \mathcal{H}^{T_1'T_2'\bar{T}_1\bar{T}_2Q_2'F_I} \to \mathcal{H}^{A_OB_OC_O\bar{Y}\bar{Z}}$ (where $Y, Z, \bar{Y}$ and $\bar{Z}$ are appropriate

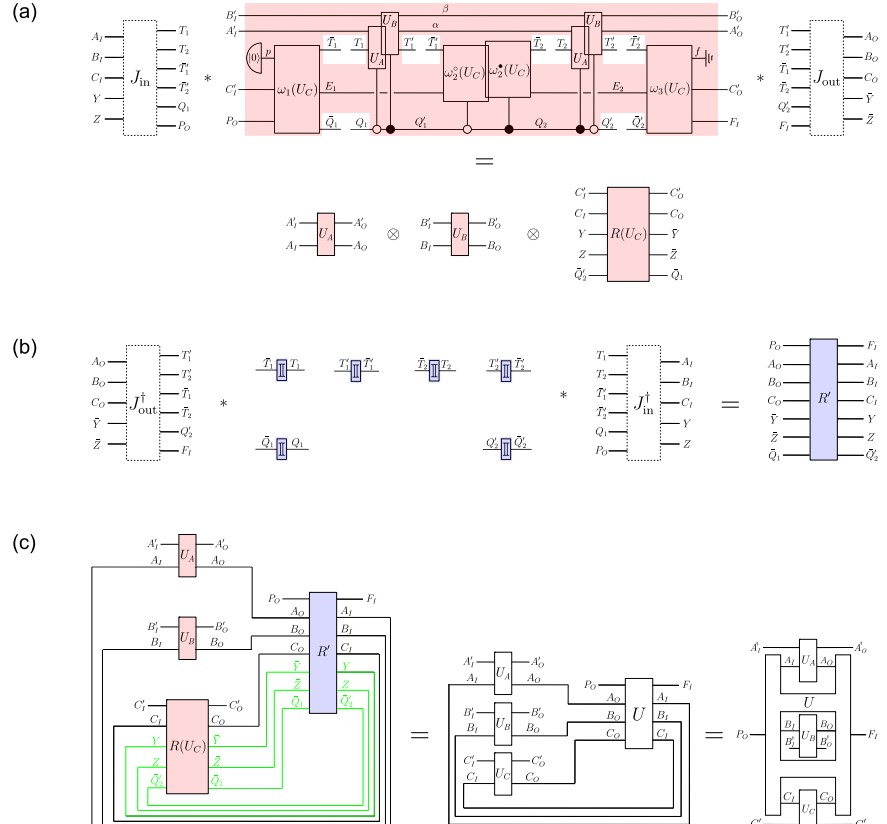

**Fig. 4 | Description of the tripartite temporal circuit in terms of time-delocalised subsystems. a** Description of the red circuit fragment in terms of the time-delocalised subsystems $A_I, B_I, C_I, Y, Z$ of the joint system $T_1 T_2 \bar{T}'_1 \bar{T}'_2 Q_1 P_O$, and $A_O, B_O, C_O, \bar{Y}, \bar{Z}$ of the joint system $T'_1 T'_2 \bar{T}_1 \bar{T}_2 Q'_2 F_I$. **b** Description of the blue circuit fragment in terms of the time-delocalised subsystems $A_I, B_I, C_I, Y, Z$ of the joint system $T_1 T_2 \bar{T}'_1 \bar{T}'_2 Q_1 P_O$, and $A_O, B_O, C_O, \bar{Y}, \bar{Z}$ of the joint system $T'_1 T'_2 \bar{T}_1 \bar{T}_2 Q'_2 F_I$. **c** The composition of the operations $R(U_C)$ and $R'$ over the systems $Y, \bar{Y}, Z, \bar{Z}, Q_1, \bar{Q}'_2$ gives

rise to a cyclic circuit fragment consisting of the operation $U_C$ and the unitary $U$ that defines the process. That is, when evaluating the composition of $R(U_C)$ and $R$ over the wires shown in green (but not over $C_I$ and $C_O$), we obtain the cyclic circuit in the middle, consisting of the operations $U_A, U_B, U_C$ and $U$. With respect to the systems $P_O, A'_{IO}, B'_{IO}, C'_{IO}, F_I$, as well as the time-delocalised systems $A_{IO}, B_{IO}, C_{IO}$, the circuit therefore consists of $U_A, U_B, U_C$ and $U$, composed in a cyclic manner as in the process matrix framework.

complementary subsystems) are defined in Supplementary Note 3B, based on a specific decomposition of unitarily extended process vectors which plays a central role in the bipartite proof (cf. Supplementary Equation (3)), and which generalises to the multipartite case (cf. Supplementary Equation (24)).

In Supplementary Note 3C, we change to the description of the circuit in terms of the corresponding time-delocalised subsystems. For that purpose, we decompose the circuit into the red and blue circuit fragment shown in Fig. 4. By construction, when composed with $J_{in}$ and $J_{out}$, the red circuit fragment shown in Fig. 4a consists of precisely one application of $U_A$ and $U_B$, in parallel to a unitary operation $R(U_C) : \mathcal{H}^{C_I C_I Y Z \bar{Q}'_2} \rightarrow \mathcal{H}^{C_O C_O \bar{Y} \bar{Z} Q_1}$. Under that change of subsystems, the complementary blue fragment needs to be composed with the inverse isomorphisms $J^\dagger_{in}$ and $J^\dagger_{out}$, which results in an operation $R' : \mathcal{H}^{P_O A_O B_O C_O \bar{Y} \bar{Z} Q_1} \rightarrow \mathcal{H}^{F_I A_I B_I C_I Y Z \bar{Q}'_2}$ (Fig. 4b). $R(U_C)$ and $R'$ cannot be further decomposed for now.

At this point, we thus have a cyclic circuit which consists of the four boxes $U_A, U_B, R(U_C)$ and $R'$, and which involves the systems $P_O, A^{(')}_{IO}, B^{(')}_{IO}, C^{(')}_{IO}, F_I$, as well as $Y, \bar{Y}, Z, \bar{Z}, Q_1, \bar{Q}'_2$ (see the left-hand side of Fig. 4c). In order to obtain a description with respect to only the systems $P_O, A^{(')}_{IO}, B^{(')}_{IO}, C^{(')}_{IO}, F_I$, we need to evaluate the composition of $R(U_C)$ and $R'$ over the systems $Y, \bar{Y}, Z, \bar{Z}, Q_1, \bar{Q}'_2$ (but not over the systems $C_I$ and $C_O$, which we wish to maintain in the description). The isomorphisms $J_{in}$ and $J_{out}$ are constructed in precisely such a way (based on the abstract relation between the systems in the process that is also used in the

bipartite proof) that, when this composition of $R(U_C)$ and $R'$ over $Y, \bar{Y}, Z, \bar{Z}, Q_1, \bar{Q}'_2$ is evaluated, the result is a cyclic circuit fragment consisting of the unitary operation $U : \mathcal{H}^{P_O A_O B_O C_O} \rightarrow \mathcal{H}^{F_I A_I B_I C_I}$ that defines the process, composed with the operation $U_C : \mathcal{H}^{C_I C_I} \rightarrow \mathcal{H}^{C_O C_O}$ (see the middle of Fig. 4c). (Note the particularity that $U_C$ only appears as an explicit part of the cyclic circuit after this composition of $R(U_C)$ with $R'$, and is not a tensor product factor of $R(U_C)$).

Therefore, in its description with respect to the systems $P_O, A^{(')}_{IO}, B^{(')}_{IO}, C^{(')}_{IO}, F_I$, the circuit in Fig. 3 indeed consists of the four operations $U_A : \mathcal{H}^{A_I A'_I} \rightarrow \mathcal{H}^{A_O A'_O}, U_B : \mathcal{H}^{B_I B'_I} \rightarrow \mathcal{H}^{B_O B'_O}, U_C : \mathcal{H}^{C_I C'_I} \rightarrow \mathcal{H}^{C_O C'_O}$ and $U : \mathcal{H}^{P_O A_O B_O C_O} \rightarrow \mathcal{H}^{F_I A_I B_I C_I}$, connected in a cyclic circuit as in the process matrix framework (see the right-hand side of Fig. 4c). This establishes the tripartite result.

Note that a similar construction is possible when one considers an asymmetric tripartite temporal circuit where $U_A$ is applied at a given, well-defined time, and $U_B$ either before or after it, coherently depending on the control systems (or vice versa, with the roles of $A$ and $B$ exchanged).

## A process that violates causal inequalities on time-delocalised subsystems

In ref. [10], it was shown that, for three and more parties, there exist process matrices that violate causal inequalities and that can be interpreted as classical process matrices, since they are diagonal in the computational basis. An example, first found by Araújo and Feix and

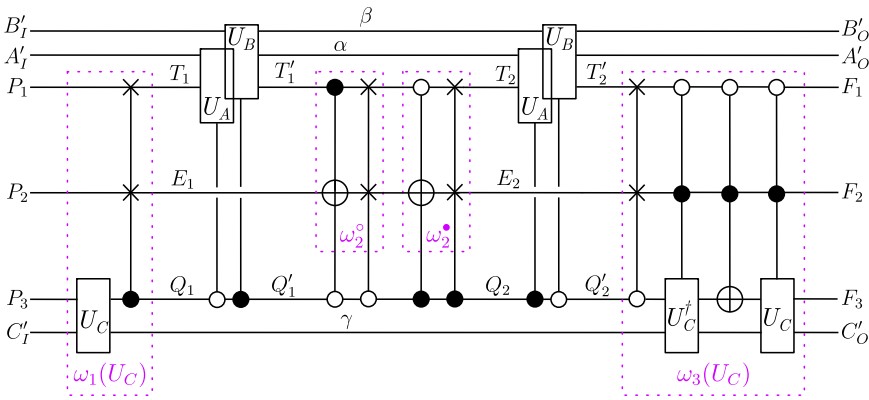

**Fig. 5 | Realisation on time-delocalised subsystems of $|U_{BW}\rangle\rangle$.** Note that for this particular process, the two circuit operations $\omega_2^{\circ}$ and $\omega_2^{\bullet}$ do not depend on $U_C$. For simplicity of the representation, the identity channels $\mathbb{1}^{T_1 \to T_1}, \mathbb{1}^{T_1' \to \bar{T}_1'}, \mathbb{1}^{\bar{T}_2 \to T_2}$, $\mathbb{1}^{T_2 \to \bar{T}_2}, \mathbb{1}^{Q_1 \to Q_1}$ and $\mathbb{1}^{Q_2 \to \bar{Q}_2}$ that constitute the blue circuit fragment in Fig. 4 are omitted in the figure here. Note that, with respect to the general tripartite circuit of Fig. 3, we can make a few simplifications for this particular process. In order to match the general form, the ancilla $\gamma$ would need to be incorporated into the circuit ancillas $E_1$, and $E_2$. But since it is just transmitted identically from $\omega_1(U_C)$ to $\omega_3(U_C)$, we may keep it as a separate wire. We can also omit the additional systems $p$ and $f$, which we introduce in Supplementary Note 3A in order to derive an alternative temporal circuit (Supplementary Fig. 5) for general unitarily extended bipartite processes (and from which we then obtain the circuit of Fig. 3 for general unitarily extended tripartite processes). The four circuit operations can be further broken

down into several temporal steps, as shown within the purple boxes. This allows one to get a descriptive understanding of how the time-delocalisation of Charlie's operation comes about in this realisation. Namely, a time-local instance of $U_C$ is applied once as part of the first circuit operation, and determines the state of the control systems that determine coherently whether $U_A$ is applied on $T_1^{()}$ and $U_B$ on $T_2^{()}$ or vice versa (i.e., their order). After they have both been applied, a reversal and reapplication of $U_C$ may occur, with a NOT gate in between, and whether this happens or not is determined jointly (and coherently, again) by the operations of Alice and Bob. However, we emphasize again that the occurence of several time-local operations that depend on $U_C$ should not be interpreted as $U_C$ being applied multiple times. Instead, just like $U_A$ and $U_B$, it is applied once and only once, on time-delocalised input and output systems.

further studied by Baumeler and Wolf in refs. [11,42], is the process matrix

$$W_{AF} = \sum_{a_O b_O c_O} |\neg b_O \wedge c_O, \neg c_O \wedge a_O, \neg a_O \wedge b_O\rangle\langle \neg b_O \wedge c_O, \neg c_O \wedge a_O, \neg a_O \wedge b_O|^{A_I B_I C_I}$$
$$\otimes |a_O, b_O, c_O\rangle\langle a_O, b_O, c_O|^{A_O B_O C_O}, \tag{4}$$

where $a_O, b_O, c_O \in \{0, 1\}$ and where $\neg$ is the negation. It was then shown by Baumeler and Wolf[42] (cf. also refs. [34,43]) that $W_{AF}$ has a unitary extension $W_{BW} = |U_{BW}\rangle\rangle\langle\langle U_{BW}|$, with

$$|U_{BW}\rangle\rangle = \sum_{\substack{a_O b_O c_O \\ p_1 p_2 p_3}} |p_1, p_2, p_3\rangle^{P_1 P_2 P_3} \otimes |p_1 \oplus \neg b_O \wedge c_O, p_2 \oplus \neg c_O \wedge a_O, p_3 \oplus \neg a_O \wedge b_O\rangle^{A_I B_I C_I}$$
$$\otimes |a_O, b_O, c_O\rangle^{A_O B_O C_O} \otimes |a_O, b_O, c_O\rangle^{F_1 F_2 F_3} \tag{5}$$

(with $p_1, p_2, p_3 \in \{0, 1\}$, i.e., $\mathcal{H}^{P_O} = \mathcal{H}^{P_1 P_2 P_3}$ and $\mathcal{H}^{F_I} = \mathcal{H}^{F_1 F_2 F_3}$ consisting of three qubits each, and with $\oplus$ denoting addition modulo 2). $W_{AF}$ is recovered from $|U_{BW}\rangle\rangle\langle\langle U_{BW}|$ when the global past party prepares the state $|0, 0, 0\rangle\langle 0, 0, 0|^{P_1 P_2 P_3}$, and the global future party is traced out. What kind of temporal circuit do we obtain when we apply the general tripartite considerations from the previous section to this particular example? A possible such realisation of this process on time-delocalised subsystems is given by the circuit shown in Fig. 5 (similar circuits corresponding to this process have also been studied in other contexts in refs. [43–45]).

In Supplementary Note 4A, we give the explicit expressions of the circuit operations in Fig. 5, as well as for the isomorphisms that define the description in terms of time-delocalised subsystems for this particular case, and we sketch how to apply the general tripartite proof to this example.

The abstract process $W_{AF}$ in Eq. (4) violates causal inequalities when each party performs a computational basis measurement on its incoming Hilbert space (and outputs the measurement result $o_X$), and prepares the computational basis state $|i_X\rangle$ (corresponding to its classical input $i_X$) on its outgoing Hilbert space. The corresponding

unitary operations that need to be applied in the pure description of the process (and therefore in the circuit of Fig. 5) are $U_X = \mathbb{1}^{X_I \to X_O} \otimes \mathbb{1}^{X_I' \to X_O}$, with each incoming ancillary system being prepared in the state $|i_X\rangle^{X_I'}$ and the outgoing ancillary systems being measured in the computational bases. One obtains the deterministic correlation $P(o_A, o_B, o_C|i_A, i_B, i_C) = \delta_{o_A, \neg i_B \wedge i_C} \delta_{o_B, \neg i_C \wedge i_A} \delta_{o_C, \neg i_A \wedge i_B}$, which was shown to violate causal inequalities in ref. [11].

An example of a causal inequality that is violated by this correlation is

$$P(0,0,0|0,0,1) + P(0,0,1|0,0,1) + P(0,0,0|1,0,0) + P(1,0,0|1,0,0)$$
$$+ P(0,0,0|0,1,0) + P(0,1,0|0,1,0) - P(0,0,0|0,0,0) =: I_1 \geq 0, \tag{6}$$

which was derived in ref. [8]. (It corresponds to Eq. (26) given there, with 0 and 1 exchanged for all inputs and outputs). Here, we find that $I_1 = -1$.

Interestingly, for that particular process with these particular local operations, all operations involved in the tripartite construction simply take computational basis states to computational basis states. These can be understood as deterministic operations between classical random variables, rather than unitary operations between quantum systems. In Supplementary Note 4B, we explain this in more detail.

All things considered, our main result is thus that there exist classical, deterministic circuits, composed of operations between time-local variables, which, when described in terms of suitable time-delocalised variables, correspond to classical, deterministic processes that violate causal inequalities.

**Noncausal correlations between time-delocalised variables**

After having established that this realisation of a noncausal process exists, we now turn to the question of what we should conclude from the fact that a causal inequality can be violated in such a situation. The general reasoning behind causal inequalities is similar to that behind Bell inequalities—one considers certain assumptions which restrict the correlations that can arise from some experiment, and their violation then implies that not all of these assumptions are satisfied. To

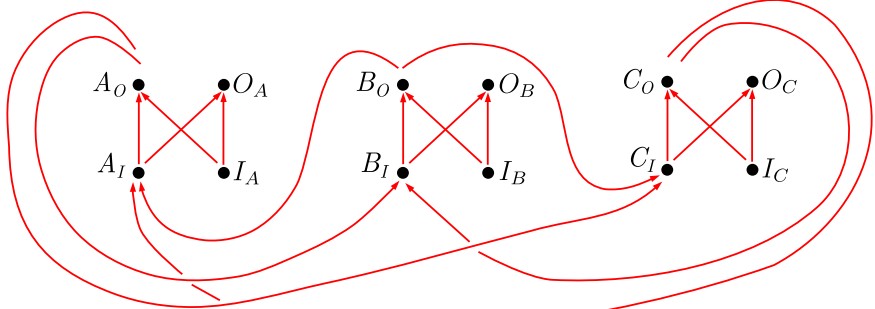

**Fig. 6 | Causal structure of the cyclic causal model corresponding to the process $W_{AF}$.** The causal influences represented by the arrows can be rigorously defined in the framework of cyclic split-node causal models[40] (or, in the case of more general quantum processes, cyclic quantum causal models[40]). In particular, if we regard each single variable as a split-node (or in the more general case of quantum processes, a quantum node where the output Hilbert space is the dual of the input Hilbert space), the experiment can be viewed as a process on a larger number of nodes, which is given by the (tensor) product of the original process and the local operations of the parties. The causal relations between the new nodes form the cyclic causal structure in Fig. 6. This follows from the known cyclic causal structure of the process $W_{AF}$[40] and the most general causal structure that each local operation from $X_I, I_X$ to $O_X, O_X, X = A, B, C$, can have. (Here, we are imagining an experiment in which each party could choose over a finite set of local operations that could instantiate all these different causal relations. The set of operations over which the party can choose can be embedded within a single deterministic operation with the maximally connected acyclic causal structure displayed in the figure by choosing $I_X$ and $O_X$ of sufficiently large cardinality.) This causal structure can be operationally verified: by applying time-delocalised SWAP operations on the time-delocalised variables (or quantum systems) so as to disconnect the operations of the parties (see Supplementary Note 1), one could intervene on the variables (or quantum systems) and verify which ones are directly influenced by which other ones. Note that the process $W_{AF}$ was first studied in the framework of cyclic causal models in ref. [40], but from the perspective of the coarse-grained split nodes defined by the pairs of variables $(X_I, X_O) \equiv X$, for $X = A, B, C$.

determine whether a causal inequality violation is a meaningful device-independent witness of causal indefiniteness, one must therefore clarify whether the assumptions underlying causal inequalities are plausible or compelling in the setting under consideration—a question that is subtle, notably in regimes of relativistic quantum information and quantum gravity[46,47], but, as it will turn out, also in the standard quantum situations we consider here. In the following, we will therefore analyse our result in this regard, and argue that causal inequalities are indeed a meaningful concept to show the absence of a definite causal order between the time-delocalised variables we identified.

In the original approach developed in ref. [2], one firstly assumes that the events involved in the experiment take place in a causal order (which, in general, can be dynamical and subject to randomness[6,8]). With respect to this causal order, there are two further assumptions that enter the derivation of causal inequalities. Firstly, the classical inputs which the parties receive are subject to free choice. Technically speaking, this means that they cannot be correlated with any properties pertaining to their causal past or elsewhere (see Methods). Secondly, the parties operate in closed laboratories. That is, intuitively speaking, they open their laboratory once to let a physical system enter, interact with it and open their laboratory once again to send out a physical system, which provides the sole means of information exchange between the local variables and the rest of the experiment. More formally, the closed laboratory assumption says that, for each party $X$, any causal influence from the setting variable $I_X$, which describes its classical input, to any other variable, except the variable $O_X$ which describes its classical outcome, has to pass through the outgoing variable $X_O$. Similarly, any causal influence to $O_X$ from any other variable except $I_X$ has to pass through $X_I$. Furthermore, $X_I$ is in the causal past of $X_O$ (see Methods). In order to clarify whether the violation of a causal inequality discovered here is meaningful and interesting, the question that we need to address is whether one would naturally expect the free choice and closed laboratory assumptions to be satisfied in our scenario with time-delocalised (classical) variables, or whether one of them is manifestly violated.

In the Methods section "Causal inequality assumptions", we formulate these assumptions, for the multipartite case, in a way that is suitable for our time-delocalised setting, namely directly in terms of the variables involved (rather than in terms of events as in ref. [2]), and

show that they indeed imply that causal inequalities must be respected. Our formulation provides a strengthening of the original derivation in ref. [2] by relaxing the closed laboratories assumption—rather than imposing that the incoming variable $X_I$ is always in the causal past of the outgoing variable $X_O$, we only require this constraint to hold for at least one particular value of the corresponding setting variable $I_X$ (see Methods). As we discuss in the following, this formulation of the assumptions is directly motivated by the observable causal relations between the variables of interest. Thus, the violation of a causal inequality in the experiment can be seen as a compelling, device-independent demonstration of the nonexistence of a possibly dynamical and random causal order between the variables.

The causal relations between the incoming and outgoing variables $X_I$ and $X_O$, as well as the setting and outcome variables $I_X$ and $O_X, X = A, B, C$, can be graphically represented by a directed graph as in Fig. 6, where the arrows describe direct causal influences.

In the causal structure in Fig. 6, the variables $I_X$ are root variables and hence they can only be correlated with other variables as a result of causal influence from them to these other variables. It is thus natural to assume the same would be true if there existed an explanation of the correlations in terms of a definite causal order, which legitimates the free choice assumption.

Regarding the closed laboratory assumption, in the graph of Fig. 6, any causal influence from $I_X$ to variables other than $O_X$ and $X_O$ is mediated, or screened off, by $X_O$. Similarly, any influence onto $O_X$ by variables other than $I_X$ and $X_I$ is mediated by $X_I$. It is natural to assume that these constraints would also hold in any potential explanation of the correlations in terms of a definite causal order. Finally, the causal diagram displays causal influence from $X_I$ to $X_O$. Note that this causal influence from $X_I$ to $X_O$ can be turned on or off depending on the value of the setting variable $I_X$. This is precisely the reason why we introduced the weakened form of the closed laboratory assumption described above, which indeed allows for $X_O$ to be inside or outside of the causal future of $X_I$, depending on the value of $I_X$.

To summarise, we have shown that there is a set of natural assumptions about the possible underlying causal orders between the variables of interest in our experiment, which are directly motivated by the observable causal relations between these variables, and which imply that the correlations in the experiment would need to respect

causal inequalities. The observable violation of a causal inequality in the experiment thus implies that an underlying causal order compatible with these assumptions cannot exist.

Are there any considerations that would lead us to drop one assumption over another in this type of experiment? In particular, could it be that, in spite of the outlined considerations about the observable causal relations, a more careful inspection of the temporal description of the experiment would reveal that it is in fact the free choice or closed laboratory assumptions that is violated, as opposed to the existence of a causal order per se? In the discussion below and in Supplementary Note 6, we analyse this question and argue that if the hypothetical causal order is expected to be imposed by spatiotemporal relations, it is the existence of causal order per se that seems violated, since the variables of interest do not admit an effective localisation in spacetime.

## Discussion

A central question in the study of quantum causality is which processes with indefinite causal order have a realisation within standard quantum theory. In order to address this question, it is first of all necessary to clarify what it means for a causally indefinite process to have a standard quantum theoretical realisation, a question that is subtle and has led to a lot of controversy. An answer to this question is provided by the concept of time-delocalised subsystems, which establishes a bridge between the standard quantum theoretical description of the scenarios under consideration and their description in the process matrix framework, in which the notion of indefinite causal order is formalised. Prior to our work, it had been known that indefinite causal order can be realised on systems that are time-delocalised in a coherently controlled manner—that is, intuitively speaking, the input and output systems of each party effectively reduce to one or another time-local system, conditionally on the state of a control quantum system. Here, we showed that this paradigm does not encompass all possibilities, and that standard quantum theory also allows for more radical ways to realise indefinite causal order processes. Notably, there exist processes that have realisations on time-delocalised subsystems and that violate causal inequalities, a feature that is generally believed to be impossible within standard (quantum) physics[14]. We analysed a concrete tripartite example, for which it turned out that the situation can entirely be understood in terms of classical variables, rather than quantum systems. There, Alice's and Bob's input and output variables are time-delocalised in a classically controlled way, while the situation for Charlie is quite different. From the point of view of the temporal description of the experiment, one time-local instance of Charlie's operation is applied in the beginning of the circuit, which may be reversed and reapplied at the end of the circuit, conditionally on the output of Alice and Bob. We then analysed this causal inequality violation with regard to the assumptions that underlie the derivation of causal inequalities, and found that the free choice and closed laboratory assumptions are not manifestly violated, which makes causal inequalities a meaningful device-independent concept to qualify these realisations as incompatible with a definite causal order.

Let us further elaborate on the subtleties that this analysis involves, in particular with respect to the closed laboratory assumption (see a more detailed discussion in Supplementary Note 6). From an intuitive reading of the circuit in Fig. 5, one may be tempted to say that Charlie acts multiple times or receives several inputs, and sends out several outputs. At first sight, this seems to violate the closed laboratory assumption, which essentially stipulates that each party is involved in a single round of information exchange, where they receive information about the past through the input variable $X_I$ and subsequently send out information into the future through the output variable $X_O$. However, it is crucial to realise that the causal inequality assumptions concern concrete variables (or quantum systems), which in our case we have explicitly specified, and which are not the same as what one might intuitively assume if one thinks of this experiment as involving three laboratories existing through time that exchange information with each other. In particular, the parties Alice, Bob and Charlie must be understood abstractly as agents who control the parameters that describe the operations taking place on the time-delocalised variables. As such, they indeed apply their operations once and only once on the pairs of input and output variables we have identified. To say that the closed laboratory assumption is violated, one would need to come up with an account for the process in terms of variables which are embedded into a causal order, but for which the closed laboratory assumption fails. We are not aware of any explanation in terms of the time-local variables in the temporal circuit and the causal order defined by their spatiotemporal relations (or any other operationally meaningful variables) where this is the case. In particular, the above-outlined intuitive reading of the circuit, with the operations being effectively localised in time, conditioned on other variables in the process, while meaningful for quantumly controlled time-delocalised operations, does not make operational sense in our case (as it would mean that some future parties can influence what has happened in the past, see Supplementary Note 6). In Supplementary Note 6, we show that, for some of the time-delocalised variables we identified, there do not exist time-local variables that take their value, meaning that they do not admit any effective localisation in time.

The further implications of this finding are yet to be unravelled, and raise various open questions. In a more general sense, there is a causal explanation for how these correlations in our process come about-namely, precisely the tripartite circuit realisation we found. This raises the question of whether and how the concept of causal inequalities in itself could be revised or modified. For instance, could there be a notion of causal process which is more relaxed, and which includes such possibilities?

What other processes beyond the classes considered here have a realisation on time-delocalised subsystems, and what other types of time-delocalisation would this involve? Could it be that any indefinite causal order process admits such a realisation, or are there counter-examples? The proof for unitarily extended tripartite processes is crucially based on the fact that the bipartite unitarily extended process resulting from fixing one of the operations has a particular standard form—namely, a variation of the quantum switch[40,41]. Establishing whether a similar standard form exists for unitarily extended processes with more than two parties could give insight into whether the constructions presented here can be generalised to more parties.

Note that there are also unitary extensions of bipartite processes— i.e., variations of the quantum switch—that have realisations of the kind considered here, with one of the operations being reversed and reapplied (for instance, one obtains such a realisation when one fixes Alice's or Bob's operation in the circuit of Fig. 5). This raises the question of whether, conversely, the process considered in this work could have an alternative, more intuitive interpretation as a superposition of processes with different definite causal orders in some sense (although it cannot be achieved by direct multipartite generalisations of the quantum switch[13]). The decomposition of this process into a direct sum of causal unitary processes shown in[40] may offer insights into this question.

Finally, in the way the process framework was originally conceived, the operations performed by the parties were imagined to be local from the point of view of some local notion of time for each party. Can we conceive of a notion of a quantum temporal reference frame with respect to which the time-delocalised variables considered here would look local, and what implications would this have for our understanding of the spacetime causal structure in which these experiments are embedded? In view of the fact that the example considered here is purely classical, the question arises of which part of a noncausal process is actually related to the quantumness of causal relations. On the practical side, an obvious question is whether our

finding could unveil new applications. For instance, could we use such time-delocalised variables for new cryptographic or other information-processing protocols?

## Methods

### The Choi isomorphism and the link product

The Choi isomorphism[48] is a convenient way to represent linear maps between vector spaces as vectors themselves, and linear maps between spaces of operators as operators themselves. In order to define it, we choose for each Hilbert space $\mathcal{H}^Y$ a fixed orthonormal, so-called computational basis $\{|i\rangle\}_i$. For a Hilbert space $\mathcal{H}^{YZ} = \mathcal{H}^Y \otimes \mathcal{H}^Z$, with computational bases $\{|i\rangle^Y\}_i$ of $\mathcal{H}^Y$ and $\{|j\rangle^Z\}_j$ of $\mathcal{H}^Z$, respectively, the computational basis is taken to be $\{|i,j\rangle^{YZ} := |i\rangle^Y \otimes |j\rangle^Z\}_{ij}$. We then define the pure Choi representation of a linear operator $V : \mathcal{H}^Y \to \mathcal{H}^Z$ as

$$|V\rangle\rangle := \mathbb{1} \otimes V|\mathbb{1}\rangle\rangle^{YY} = \sum_i |i\rangle^Y \otimes V|i\rangle^Y \in \mathcal{H}^Y \otimes \mathcal{H}^Z, \quad (7)$$

with $|\mathbb{1}\rangle\rangle^{YY} := \sum_i |i\rangle^Y \otimes |i\rangle^Y \in \mathcal{H}^Y \otimes \mathcal{H}^Y$. Similarly, we define the (mixed) Choi representation of a linear map $\mathcal{M} : \mathcal{L}(\mathcal{H}^Y) \to \mathcal{L}(\mathcal{H}^Z)$ as

$$M := (\mathcal{I}^Y \otimes \mathcal{M})(|\mathbb{1}\rangle\rangle\langle\langle\mathbb{1}|^{YY}) = \sum_{i,i'} |i\rangle\langle i'|^Y \otimes \mathcal{M}(|i\rangle\langle i'|^Y) \in \mathcal{L}(\mathcal{H}^{YZ}) \quad (8)$$

where $\mathcal{I}^Y$ denotes the identity map on $\mathcal{L}(\mathcal{H}^Y)$.

The link product[32,33] is a tool which allows one to compute the Choi representation of a composition of maps in terms of the Choi representation of the individual maps. Consider two tensor product Hilbert spaces $\mathcal{H}^{XY} = \mathcal{H}^X \otimes \mathcal{H}^Y$ and $\mathcal{H}^{YZ} = \mathcal{H}^Y \otimes \mathcal{H}^Z$ which share the same (possibly trivial) space factor $\mathcal{H}^Y$, and with non-overlapping $\mathcal{H}^X, \mathcal{H}^Z$. The link product of any two vectors $|a\rangle \in \mathcal{H}^{XY}$ and $|b\rangle \in \mathcal{H}^{YZ}$ is defined (with respect to the computational basis $\{|i\rangle^Y\}_i$ of $\mathcal{H}^Y$) as[13]

$$|a\rangle * |b\rangle := (\mathbb{1}^{XZ} \otimes \langle\langle\mathbb{1}|^{YY})(|a\rangle \otimes |b\rangle) = \sum_i |a_i\rangle^X \otimes |b_i\rangle^Z \in \mathcal{H}^{XZ} \quad (9)$$

with $|a_i\rangle^X := (\mathbb{1}^X \otimes \langle i|^Y)|a\rangle \in \mathcal{H}^X$ and $|b_i\rangle^Z := (\langle i|^Y \otimes \mathbb{1}^Z)|b\rangle \in \mathcal{H}^Z$. Similarly, the link product of any two operators $A \in \mathcal{L}(\mathcal{H}^{XY})$ and $B \in \mathcal{L}(\mathcal{H}^{YZ})$ is defined as[32,33]

$$A * B := (\mathbb{1}^{XZ} \otimes \langle\langle\mathbb{1}|^{YY})(A \otimes B)(\mathbb{1}^{XZ} \otimes |\mathbb{1}\rangle\rangle)^{YY} = \sum_{ii'} A_{ii'}^X \otimes B_{ii'}^Z \in \mathcal{L}(\mathcal{H}^{XZ}) \quad (10)$$

with $A_{ii'}^X := (\mathbb{1}^X \otimes \langle i|^Y)A(\mathbb{1}^X \otimes |i'\rangle^Y) \in \mathcal{L}(\mathcal{H}^X)$ and $B_{ii'}^Z := (\langle i|^Y \otimes \mathbb{1}^Z)A(|i'\rangle^Y \otimes \mathbb{1}^Z) \in \mathcal{L}(\mathcal{H}^Z)$.

The link products thus defined are commutative (up to a re-ordering of the tensor products), and associative provided that each constituent Hilbert space appears at most twice[13,33]. For $|a\rangle \in \mathcal{H}^X$ and $|b\rangle \in \mathcal{H}^Z$, or $A \in \mathcal{L}(\mathcal{H}^X)$ and $B \in \mathcal{L}(\mathcal{H}^Z)$ in distinct, non-overlapping spaces, they reduce to tensor products $(|a\rangle * |b\rangle = |a\rangle \otimes |b\rangle$ or $A * B = A \otimes B)$. For $|a\rangle, |b\rangle \in \mathcal{H}^Y$, or $A, B \in \mathcal{L}(\mathcal{H}^Y)$ in the same spaces, they reduce to scalar products $(|a\rangle * |b\rangle = \sum_i \langle i|a\rangle\langle i|b\rangle = |a\rangle^T |b\rangle$ or $A * B = \text{Tr}[A^T B])$.

For two linear operators $V_1 : \mathcal{H}^X \to \mathcal{H}^{X'Y}$ and $V_2 : \mathcal{H}^{YZ} \to \mathcal{H}^{Z'}$, the pure Choi representation of the composition $V := (\mathbb{1}^{X'} \otimes V_2)(V_1 \otimes \mathbb{1}^Z) : \mathcal{H}^{XZ} \to \mathcal{H}^{X'Z'}$ is obtained, in terms of the pure Choi representations $|V_1\rangle\rangle \in \mathcal{H}^{XX'Y}$ and $|V_2\rangle\rangle \in \mathcal{H}^{YZZ'}$ of the individual operators $V_1$ and $V_2$, as

$$|V\rangle\rangle = |V_1\rangle\rangle * |V_2\rangle\rangle \in \mathcal{H}^{XX'ZZ'}. \quad (11)$$

Similarly, for two linear maps $\mathcal{M}_1 : \mathcal{L}(\mathcal{H}^X) \to \mathcal{L}(\mathcal{H}^{X'Y})$ and $\mathcal{M}_2 : \mathcal{L}(\mathcal{H}^{YZ}) \to \mathcal{L}(\mathcal{H}^{Z'})$ the Choi representation of the composition $\mathcal{M} := (\mathcal{I}^{X'} \otimes \mathcal{M}_2) \circ (\mathcal{M}_1 \otimes \mathcal{I}^Z) : \mathcal{L}(\mathcal{H}^{XZ}) \to \mathcal{L}(\mathcal{H}^{X'Z'})$ is obtained, in terms of the Choi representations of the individual maps $M_1 \in \mathcal{L}(\mathcal{H}^{XX'Y})$ and $M_2 \in \mathcal{L}(\mathcal{H}^{YZZ'})$ of $\mathcal{M}_1$ and $\mathcal{M}_2$, as

$$M = M_1 * M_2 \in \mathcal{L}\left(\mathcal{H}^{XX'ZZ'}\right). \quad (12)$$

Another property of the link product, which can easily be verified from its definition, is that for any $|a\rangle \in \mathcal{H}^{XY}, |b\rangle \in \mathcal{H}^{YZ}$ and any unitary $U : \mathcal{H}^Y \to \mathcal{H}^{Y'}$, it holds that

$$(|a\rangle * |U\rangle\rangle) * (|U^\dagger\rangle\rangle * |b\rangle) = |a\rangle * |b\rangle. \quad (13)$$

Similarly, for any $A \in \mathcal{L}(\mathcal{H}^{XY}), B \in \mathcal{L}(\mathcal{H}^{YZ})$ and any unitary $U : \mathcal{H}^Y \to \mathcal{H}^{Y'}$, it holds that

$$(A * |U\rangle\rangle\langle\langle U|) * (|U^\dagger\rangle\rangle\langle\langle U^\dagger| * B) = A * B. \quad (14)$$

This is precisely the property we use in the main text when changing the subsystem description of a circuit. Namely, it is due to this property that the overall composition of two circuit fragments remains the same when we compose one fragment with certain isomorphisms (i.e., unitary transformations) defining new subsystems, and the complementary fragment with the inverses of these isomorphisms.

### Unitary extensions of bipartite processes on time-delocalised subsystems

In summary, the bipartite result says that for any unitarily extended bipartite process, there exists a temporally ordered quantum circuit, with operations that depend on the local operations $U_A$ and $U_B$ applied in the process, which precisely corresponds to the situation considered in the process matrix framework, with one instance of each $U_A$ and $U_B$ composed with the process matrix in a cyclic circuit, when described in terms of a suitable choice of time-delocalised subsystems.

Formally, the bipartite result can be stated as follows.

**Proposition 2.** Consider a unitary extension of a bipartite process, described by a process vector $|U\rangle\rangle \in \mathcal{H}^{P_O A_I O B_I O F_I}$, composed with unitary local operations $U_A : \mathcal{H}^{A_I A_I'} \to \mathcal{H}^{A_O A_O'}$ and $U_B : \mathcal{H}^{B_I B_I'} \to \mathcal{H}^{B_O B_O'}$. For any such process, the following exist.

1. A temporal circuit as in Fig. 7, in which $U_A$ is applied on some systems $A_I$ and $A_O$ at a definite time, preceded and succeeded respectively by two unitary circuit operations $\omega_1(U_B) : \mathcal{H}^{B_I P_O} \to \mathcal{H}^{A_I E}$ and $\omega_2(U_B) : \mathcal{H}^{A_O E} \to \mathcal{H}^{B_O F_I}$ that depend on $U_B$.
2. Isomorphisms $J_{\text{in}} : \mathcal{H}^{B_I Z} \to \mathcal{H}^{A_O P_O}$ and $J_{\text{out}} : \mathcal{H}^{A_I F_I} \to \mathcal{H}^{B_O Z}$, such that, with respect to the subsystem $B_I$ of $A_O P_O$ and the subsystem $B_O$ of $A_I F_I$ that these isomorphisms define, the circuit in Fig. 7 takes the form of a cyclic circuit composed of $U, U_A$ and $U_B$, as in the process matrix framework (Fig. 8).

Here, we outline the main points of the proof. All technical details and calculations are given in Supplementary Note 2.

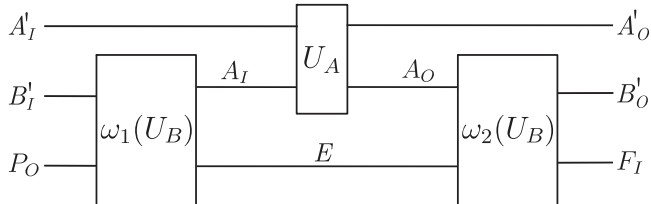

**Fig. 7 | Temporal circuit in the bipartite case.** Temporal circuit for a bipartite unitary process, with $U_A$ being applied on time-local systems $A_I$ and $A_O$, and with circuit operations $\omega_1(U_B) : \mathcal{H}^{B_I P_O} \to \mathcal{H}^{A_I E}$ and $\omega_2(U_B) : \mathcal{H}^{A_O E} \to \mathcal{H}^{B_O F_I}$ that depend on $U_B$, and that are connected by an ancillary system $E$.

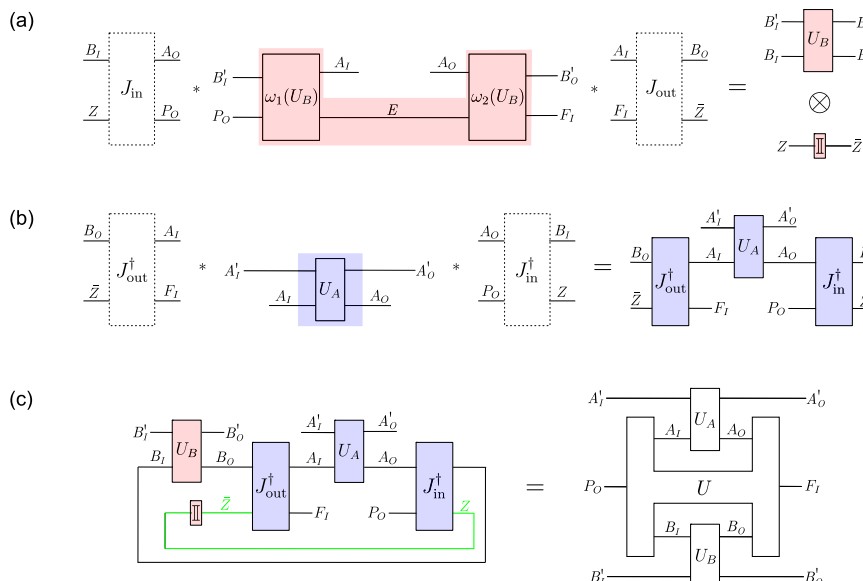

**Fig. 8 | Description of the bipartite temporal circuit in terms of time-delocalised subsystems. a** Description of the red circuit fragment, which implements an operation from $\mathcal{H}^{B_I P_O A_O}$ to $\mathcal{H}^{B_O A_I F_I}$, in terms of the time-delocalised subsystems $B_I$, $Z$ of the joint system $A_O P_O$ and $B_O$, $\bar{Z}$ of $A_I F_I$. **b** Description of the blue circuit fragment, which is simply the operation $U_A$, in terms of the time-delocalised subsystems $B_I$, $Z$ of $A_O P_O$ and $B_O$, $\bar{Z}$ of $A_I F_I$. **c** In the new subsystem description, one obtains a cyclic circuit, as considered in the process matrix framework, where the unitary operation $U$ that defines the process is obtained by composing the inverse isomorphisms $J_{\text{in}}^{\dagger}$ and $J_{\text{out}}^{\dagger}$ and the identity channel $\mathbb{1}^{Z \to \bar{Z}}$ over the subsystems $Z$ and $\bar{Z}$ (i.e., over the wires shown in green).

**Outline of proof**. The existence of a temporal circuit with the form of Fig. 7 is shown in Supplementary Note 2A. It follows from the fact that any unitary extension of a one-party process can be implemented as a fixed-order circuit or quantum comb[32,33], in which the party applies its operation at a definite time. For a unitary extension of a bipartite process, one can therefore find a fixed-order circuit in which one of the parties acts at a definite time, and which is composed of circuit operations that depend on the operation of the other party.

In Supplementary Note 2B, we show that the unitary $U$ which defines the process isomorphically maps some subsystem of $A_O P_O$ to $B_I$, and $B_O$ to some subsystem of $A_I F_I$. The corresponding isomorphisms $J_{\text{in}} : \mathcal{H}^{B_I Z} \to \mathcal{H}^{A_O P_O}$ and $J_{\text{out}} : \mathcal{H}^{A_I F_I} \to \mathcal{H}^{B_O \bar{Z}}$ (where $Z$ and $\bar{Z}$ are appropriate complementary subsystems) can be taken to define an alternative description of the circuit in Fig. 7 in terms of time-delocalised subsystems, since there, $P_O$, $A_I$, $A_O$ and $F_I$ are time-local wires.

In Supplementary Note 2C, we change to the description of the circuit in terms of these time-delocalised subsystems. For that purpose, we decompose the circuit into the red and blue circuit fragment shown in Fig. 8. By construction, when composed with $J_{\text{in}}$ and $J_{\text{out}}$, the red fragment consists of precisely one application of $U_B : \mathcal{H}^{B_I B_I'} \to \mathcal{H}^{B_O B_O'}$, in parallel to an identity channel from $Z$ to $\bar{Z}$ (Fig. 8a). The blue fragment, which is just the operation $U_A$, needs to be composed with the inverse isomorphisms $J_{\text{in}}^{\dagger}$ and $J_{\text{out}}^{\dagger}$ so that the overall, global transformation implemented by the circuit remains the same (Fig. 8b). In the new description of the circuit of Fig. 7 in terms of these subsystems, one thus obtains a cyclic circuit as on the left-hand side of Fig. 8c).

The final step is to note that the composition of the inverse isomorphisms $J_{\text{in}}^{\dagger}$ and $J_{\text{out}}^{\dagger}$ with the identity channel $\mathbb{1}^{Z \to \bar{Z}}$ over the systems $Z$ and $\bar{Z}$ is precisely the unitary operation $U$ that defines the process. Therefore, in this coarse-grained description with respect to the systems $P_O$, $A_{IO}^{(\prime)}$, $B_{IO}^{(\prime)}$, and $F_I$, the circuit indeed consists of three transformations $U_A : \mathcal{H}^{A_I A_I'} \to \mathcal{H}^{A_O A_O'}$, $U_B : \mathcal{H}^{B_I B_I'} \to \mathcal{H}^{B_O B_O'}$ and $U : \mathcal{H}^{P_O A_O B_O} \to \mathcal{H}^{F_I A_I B_I}$ that are composed in a cyclic circuit as in the process matrix picture (see the right-hand side of Fig. 8c). In other words,

it is precisely that structure that happens on the subsystems with respect to which we chose to describe the circuit. This establishes the bipartite result.

Applying the bipartite constructions presented here to the particular case of the quantum switch leads to an asymmetric implementation with Alice performing a time-local operation and Bob's operation being time-delocalised through coherent control of the times at which it is applied. For symmetric implementations in which both Alice's and Bob's operation are time-delocalised, a similar argument can be made[29].

## Causal inequality assumptions

A causal order between the elements of some set $\mathcal{S}$ is formally described by a strict partial order (SPO) on $\mathcal{S}$[2,6]. A SPO is a binary relation $\prec$, which, for all $X, Y, Z \in \mathcal{S}$, satisfies irreflexivity (not $X \prec X$) and transitivity (if $X \prec Y$ and $Y \prec Z$, then $X \prec Z$). (Note that irreflexivity and transitivity together imply asymmetry, i.e., if $X \prec Y$, then not $Y \prec X$.) If $X \prec Y$, we will say that $X$ is in the causal past of $Y$ (equivalently, $Y$ is in the causal future of $X$). For $X \neq Y$ and not $X \prec Y$, we will use the notation $X \nprec Y$, and the terminology $X$ is not in the causal past of $Y$ (equivalently, $Y$ is not in the causal future of $X$). If $X \nprec Y$ and $Y \nprec X$, we will say that $X$ is in the causal elsewhere of $Y$[49] (sometimes also termed $X$ is not causally connected to $Y$, or $X$ is causally disconnected from $Y$). For subsets $\mathcal{S}' \subset \mathcal{S}$, we will use the short-hand notation $X \preceq \mathcal{S}'$ to denote that $\forall Y \in \mathcal{S}', X \preceq Y$. We furthermore define the causal past of $X$ as the set $\mathcal{P}_X := \{Y \in \mathcal{S} | Y \prec X\}$, the causal future of $X$ as $\mathcal{F}_X := \{Y \in \mathcal{S} | X \prec Y\}$ and the causal elsewhere of $X$ as $\mathcal{E}_X := \{Y \in \mathcal{S} | Y \preceq X \text{ and } X \preceq Y\}$. Also, note that a SPO on $\mathcal{S}$ naturally induces a SPO on any subset of $\mathcal{S}$.

The variables involved in the process under consideration are the time-delocalised incoming and outgoing variables $A_I, A_O, B_I, B_O, C_I, C_O$, as well as the settings and outcomes, which can be described by random variables $I_A, I_B, I_C$ (with values $i_A, i_B, i_C$, respectively) and $O_A, O_B, O_C$ (with values $o_A, o_B, o_C$, respectively). We will abbreviate the set of all these variables to $\Gamma := \{A_I, A_O, B_I, B_O, C_I, C_O, I_A, O_A, I_B, O_B, I_C, O_C\}$. The assumption that the correlations $P(o_A, o_B, o_C | i_A, i_B, i_C)$ arise from a situation in which these variables occur in a (generally probabilistic and dynamical) causal order can be formalised as follows.

*Causal order assumption*. There exists a random variable which takes values $\kappa(\Gamma)$ in the possible strict partial orders on the set $\Gamma$, and a joint probability distribution $P(o_A, o_B, o_C, \kappa(\Gamma)|i_A, i_B, i_C)$, which, when marginalised over that variable, yields the correlations $P(o_A, o_B, o_C, |i_A, i_B, i_C)$ observable in the process, i.e.,

$$\sum_{\kappa(\Gamma)} P(o_A, o_B, o_C, \kappa(\Gamma)|i_A, i_B, i_C) = P(o_A, o_B, o_C, |i_A, i_B, i_C). \quad (15)$$

This probability distribution satisfies the following two conditions.

1. *Free choice*. The settings $I_A, I_B$ and $I_C$ are assumed to be freely chosen, which means that they cannot be correlated with any properties pertaining to their causal past or elsewhere. That is, the probability for their causal past and elsewhere to consist of certain variables, for the variables in these sets to have a certain causal order, and for the outcome variables in these sets to take certain values, cannot depend on the respective setting. Formally, with respect to $I_A$, for any (disjoint) subsets $\mathcal{Y}$ and $\mathcal{Z}$ of $\Gamma \setminus \{I_A\}$, and any causal order $\kappa(\mathcal{Y} \cup \mathcal{Z})$ on the variables in $\mathcal{Y} \cup \mathcal{Z}$, the following must hold:

$$P(o^{\mathcal{Y}}, o^{\mathcal{Z}}, \mathcal{P}_{I_A} = \mathcal{Y}, \mathcal{E}_{I_A} = \mathcal{Z}, \kappa(\mathcal{Y} \cup \mathcal{Z})|i_A, i_B, i_C)$$
$$= P(o^{\mathcal{Y}}, o^{\mathcal{Z}}, \mathcal{P}_{I_A} = \mathcal{Y}, \mathcal{E}_{I_A} = \mathcal{Z}, \kappa(\mathcal{Y} \cup \mathcal{Z})|i_B, i_C). \quad (16)$$

Here, by $P(o^{\mathcal{Y}}, o^{\mathcal{Z}}, \mathcal{P}_{I_A} = \mathcal{Y}, \mathcal{E}_{I_A} = \mathcal{Z}, \kappa(\mathcal{Y} \cup \mathcal{Z})|i_A, i_B, i_C)$, we denote the probability that is obtained from $P(o_A, o_B, o_C, \kappa(\Gamma)|i_A, i_B, i_C)$ by marginalising over all $O_X \notin \mathcal{Y} \cup \mathcal{Z}$, and by summing over all $\kappa(\Gamma)$ that satisfy the specified constraints—that is, all $\kappa(\Gamma)$ for which the causal past $\mathcal{P}_{I_A}$ of $I_A$ is $\mathcal{Y}$, the causal elsewhere $\mathcal{E}_{I_A}$ of $I_A$ is $\mathcal{Z}$, and the causal order on the subset $\mathcal{Y} \cup \mathcal{Z}$ is $\kappa(\mathcal{Y} \cup \mathcal{Z})$. The free choice assumption is that this probability is independent of the value of $I_A$. The analogous conditions must hold with respect to $I_B$ and $I_C$.

2. *Closed laboratories*. The second constraint is the closed laboratory assumption, which says, intuitively speaking, that causal influence from $I_A$ to any other variable except $O_A$ has to pass through $A_O$; that, similarly, any causal influence to $O_A$ from any other variable except $I_A$ has to pass through $A_I$; and that $A_I$ is in the causal past of $A_O$ (and analogously for $B$ and $C$). Note that, in the original derivation of causal inequalities[2], it was assumed that $X_I \prec X_O$ always holds. Here, we weaken this assumption by requiring that this constraint only holds for at least one particular value of the corresponding setting variable $I_X$. The reason is that this weakened form of the assumption (unlike the stronger assumption of $X_I \prec X_O$ regardless of the value of $I_X$) is directly motivated by the observable causal relations in our situation with time-delocalised variables (see the discussion in the main text).

This closed laboratory assumption can be formalised as a constraint on the possible causal orders as follows.

$P(o_A, o_B, o_C, \kappa(\Gamma)|i_A, i_B, i_C) > 0$ only if $\kappa(\Gamma)$ satisfies the following properties for all $Y \in \Gamma$ : i) $I_A \prec Y$, iff $Y = O_A$ or $Y = A_O$ or $A_O \prec Y$. ii) $Y \prec O_A$, iff $Y = I_A$ or $Y = A_I$ or $Y \prec A_I$.

$$(17)$$

Furthermore, there exists at least one value $i_A^*$ of $I_A$ for which $A_I \prec A_O$ with certainty, that is

$$P(o_A, o_B, o_C, \kappa(\Gamma)|i_A^*, i_B, i_C) > 0 \quad \text{only if } \kappa(\Gamma) \text{ satisfies } A_I \prec A_O. \quad (18)$$

The analogous conditions must be satisfied for $B$ and $C$.

We show in Supplementary Note 5 that this causal order assumption—notably, even with the weakened form of the closed laboratory condition we introduced—implies that the correlations $P(o_A, o_B, o_C|i_A, i_B, i_C)$ that are established in the process must be causal[6-8]. Such correlations form a polytope, whose facets precisely define causal inequalities[6-8].

(Note furthermore that we could similarly weaken the assumption that $O_A$ is always in the causal future of $A_I$. This would however change nothing about the argument, and the proof from Supplementary Note 5 would go through in the same way).

Here, we presented the argument in the classical case for concreteness, but it can be readily extended to a quantum process, or even an abstract process[6] possibly compatible with more general operational probabilistic theories (OPTs)[37,38], where there is no analogue of the classical variables $X_I$ and $X_O$. Indeed, in the general case all elements of the argument remain the same, except that the objects $X_I$ and $X_O$ over which the partial order is assumed would be general systems rather than classical variables ($I_X$ and $O_X$ will remain classical). Moreover, the argument applies analogously for any number of parties, so we have assumptions applicable to the most general case of a process.

## Data availability
Data sharing not applicable to this article as no datasets were generated or analysed during the current study.

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

## Acknowledgements

This publication was made possible through the support of the ID# 61466 grant and ID# 62312 grant from the John Templeton Foundation, as part of the project https://www.templeton.org/grant/the-quantum-information-structure-of-spacetime-qiss-second-phase 'The Quantum Information Structure of Spacetime' (QISS). The opinions expressed in this project/publication are those of the author(s) and do not necessarily reflect the views of the John Templeton Foundation. This work was supported by the Program of Concerted Research Actions (ARC) of the Université libre de Bruxelles and by the French National Research Agency through its "Investissements d'avenir" (ANR-15-IDEX-02) program and the ANR-22-CE47-0012 project. J.W. is supported by the Chargé de Recherche fellowship of the Fonds de la Recherche Scientifique FNRS (F.R.S.-FNRS). O.O. is a Research Associate of the Fonds de la Recherche Scientifique (F.R.S.-FNRS). Published with the support of the University Foundation of Belgium.

## Author contributions

J.W., C.B., and O.O. all contributed extensively to the work presented in this paper.

## Competing interests

The authors declare no competing interests.
