## [Peer Review File · Nature Communications]

Existence of processes violating causal inequalities on time-delocalised subsystemsREVIEW OF JULIAN WECHS, CYRIL BRANCIARD, AND OGNAN ORESHKOV, *EXISTENCE OF PROCESSES VIOLATING CAUSAL INEQUALITIES ON TIME-DELOCALISED SUBSYSTEMS*

A. Qualitative evaluation

The paper *Existence of processes violating causal inequalities on time-delocalised subsystems* by J. Wechs et al. approaches the difficult problem of interpreting and implementing quantum causal structures with indefinite causality. More specifically, they consider case of purifiable processes. The field believes that only purifiable process might have a physical interpretation and realization. However, it is known that purifiable process of two agents cannot break causal inequalities, as several papers (one including one of the present authors) showed. Therefore, this paper considers the case of purifiable processes involving 3 agents. Notably, the famous Lugano-process (here referred to as AF- or BW-process) is purifiable and breaks causal inequalities. Inspired by Oreshkov's concept of *time-delocalized subsystems*, the authors develop a quasi-circuit picture for such processes.

Mathematically, the paper is very sophisticated and solid, except for a few small points that I will ask about below. The paper is guaranteed to have impact in the quantum causal modeling field. To have more general impact, the authors should provide a more intuitive interpretation of time-delocalized systems that goes beyond mathematical concepts such as isomorphisms. If one could explain to other physicists what an indefinite causal structure breaking causal inequalities "looks like", that would certainly be very impressive.

It would be particularly spectacular if the authors could propose a table-top quantum-optics experiment that implements those processes, similar to the quantum switch. While such an experiment would necessarily suffer from a similar controversy as the quantum switch experiments, this would mostly be a matter of taste.

Several formulations in the paper give the impression that the authors believe that their representation is implementable. However, currently I do not see how to implement their quasi-circuit, while using the agents' operations only once. I will explain my doubts below.

B. Technical details

1. Several times, e.g. at the end of page 2, you claim that causal inequalities are device-independent. However, the apparent device-independence is only true assuming some idealizations that may fail in crucial regimes of relativistic quantum information and quantum gravity. Since these are the regimes in which we expect indefinite causal structures to arise in nature, it is crucial to evaluate whether a causal inequality breaks down because of indefinite causal structure, or because the required idealizations fail. This, however, will require theory-specific and physics-specific testing and reasoning. Two crucial examples for these problems with device-independence of causal indefiniteness already exist in the literature. The paper

1. C. T. Marco Ho, Fabio Costa, Christina Giarmatzi, Timothy C. Ralph, *Violation of a causal inequality in a spacetime with definite causal order*, arXiv:1804.05498

shows that the problems of wave-package spreading and localization in quantum field theory can lead to the break down of causal inequalities in an ordinary definite classical background spacetime structure. Furthermore, the recent paper

2. Kacper Dębski, Magdalena Zych, Fabio Costa, Andrzej Dragan, *Indefinite temporal order without gravity*, arXiv:2205.00164

shows that the relativistic phenomenon of *entanglement harvesting* allows observers to become correlated despite them not being allowed to signal to each other, and despite there being no other common cause than the quantum vacuum which is always present.

2. Now, let me explain my doubts about the implementability of your quasi-circuits. This involves Figure 5 and Figure 6, and furthermore Figure 7 for the Lugano-process specifically. In Figure 5, U_A and U_B can be implemented via superposition of trajectories, similarly as in table-top experiments of the quantum switch, and are therefore unproblematic, up to the usual problem of fair resource counting. The problem is U_C . We have three gates ω_1 , ω_2 and ω_3 , all of which depend on U_C . Therefore, Figure 5 might use U_C three times. This means that Figure 5 does not really give insights about how to implement the processes physically, while only using each agent operation once. Figure 6 b) and Figure 6 c) use signalling to the past, so also these are not valid implementations. But Figure 6 a) might give some hints about implementations. During the proofs, it does not become too clear what precise U_C -dependence J_{in} , J_{out} and $R(U_C)$ have. Is there a chance to implement a process this way, while only using the agents' operations once? If not, I am out of ideas.

How do you think your quasi-circuits help to implement processes? If you do not believe that your schemes can be implemented without using agent operations several times, then what else is it that your quasi-circuits offer?

If your advantage is in terms of interpretation, then I would like to compare your quasi-circuits to those of Allard-Guerin and Brukner in your reference [35]. Inspired by the concept of time-delocalized systems, they offer a so-called *causal reference decomposition* of purifiable processes. These are associated with one agent (say, A), who sees a time-evolution of the form $U = U_{\text{future}}(U_B, U_C) \cdot U_A \cdot U_{\text{past}}(U_B, U_C)$. What is the advantage of your representation over theirs? Also they propose a detailed quasi-circuit for the Lugano/BW/AF-process. Similar to your Figure 7, it has the problem that at least one of the other agents' operations is used several times. In the paper, you claim that the gates are only applied once, because, U_C gets applied, and later on it gets canceled by U_C^\dagger , and at the end U_C is applied again. However, if one tried to implement this circuit gate-after-gate, then one would see the use of three gates that depend on U_C , instead of one or two.

I see two crucial advantages in your quasi-circuits over those of Allard-Guerin and Brukner. First of all, up to the issues of resource counting in superpositions of trajectories, your scheme only uses U_A and U_B once. Secondly, your scheme gives an explicit quasi-circuit for all pure three-party processes, up to the $\omega_j(U_C)$ which could in principle be recovered from the constructive proof. Do you see other advantages?

3. The general relativity literature already has a name for what you call *causal elsewhere*: One says X is not causally connected to Y , or X is causally disconnected from Y . You may consider using that term instead.
4. It is true that you *Free choice*-postulate contains the ability to do actions that were not predictable from earlier knowledge. However, your *free choice*-postulate assumes much more: It assumes no signaling to the (causal) past, and no faster-than-light signaling (i.e. no signaling to the causal "elsewhere".)

5. In your Figure 11, one needs to act on internal wires, while the bipartite case only required swaps with open wires. In particular, in the tripartite case it seems that one has to have access to more than just the agents' inputs and outputs. Does that pose problems for the operational meaning?
6. In the beginning of Appendix C 2, about U_1 and U_2 : Intuitively and physically, it makes sense to me that such maps exist. However, I cannot see a mathematical proof that such maps exist/are well-defined, or that the decomposition of Equation (C3) is possible. Do you have a proof? Or does it have to be an extra postulate (it would be a reasonable one)? Furthermore, why are U_1, U_2 unitary? Is this necessary to make U unitary?
7. In Appendix C 2: Why is $|U_G(S_A, \dot{\cdot})\rangle$ a valid process vector?
8. Can you explain in more detail why Equation (D3) is true? It looks like an expansion of the brackets leads to cross-terms like $|\nu_1^{A<B}\rangle\rangle * |\tilde{\nu}_2^{A<B}\rangle\rangle$.
9. It is not obvious to me how Equation (D3) solves the problem of different sized ancillas. Is this still a unitary process? How exactly does $|U\rangle\rangle$ get enlarged to act on the other ancillas? Why does U^{comp} help? Why is it necessary?
10. In Equation (D4), I cannot see that ω_1 is unitary.
11. Can you prove that Figure 7 indeed gives the Lugano/Baumeler-Wolf/Araujo-Feix process?
12. Appendix E seems overcomplicated. Your *free choice*-requirement also includes a condition that there is no signalling to the causal past and the causal "elsewhere". In other words, it says there can only be signalling to the causal future. Equation (8) just seems like a rephrasing of no-signalling-outside-the-causal-future, and so does Equation (E3).

Reviewer #3 (Remarks to the Author):

Existence of processes violating causal inequalities on time-delocalised subsystems.

=====

Indefinite causal orders is a new development of quantum information and computation, in which one considers superposing not just the data that goes through the quantum circuit gates, but the order of the gates itself. It has been shown at the theoretical level that these could bring communication and computational complexity gains.

Say there are three agents controlling gates A, B, C.

Mathematically the process that combines these three in a superposition of orders is described by a process matrix, of which some can be excluded on the grounds of probability conservation, leaving only the valid ones.

Some indefinite causal orders have been implemented in the lab, although the proof of indefiniteness was only device-dependent, leaving open interpretations whereby the implementation actually works by using gate A several times.

Here, the authors study time-delocalized implementations of three-partite process matrices, whereby the process is compiled into a circuit where A, B and C are actually distributed in time and space globally over the gates of the circuit.

The main contribution of the paper is a proof that valid unitary 3-partite process matrices are implementable as time-delocalized systems.

In particular, the Baumeler and Wolf unitary 3-partite process is implementable.

But the Baumeler and Wolf was proved to break the causal inequalities, i.e. the ultimate, device-independent test of indefiniteness.

Thus we have here the first proof that there can be, in nature, process breaking the causal inequality.

The deep consequences of breaking the causal inequalities are also discussed: for instance the process cannot be unfolded into histories where A is done several times.

I STRONGLY RECOMMEND PUBLISHING THS PAPER.

It contains a major advance for a topic.

It is carefully drafted.

Reviewer #4 (Remarks to the Author):

The authors present a result on the existence of a realisation on time-delocalised systems for tripartite processes that admit a unitary extension. This result nicely connects with that of the existence of tripartite processes with unitary extension that violate causal inequalities. In extremely simplified terms, one can say that the notion of time-delocalised subsystems refers to some factorization of the Hilbert spaces in which some subsystems do not correspond to the same time.

The central results of the manuscript are Propositions 1 and 2, and the construction of the time-delocalised version of the AFBW process. Propositions 1 and 2 are rather technical, so it is difficult to provide here a simplified account of them.

Proposition 1 shows the existence of a specific construction for a bipartite unitary process in terms of a localised operation for Alice and a delocalised operation for Bob (point 1). The second point, similarly, concerns some time-delocalised circuit implementation. I must say that it is rather confusing in its formulation. Of course, one can recover its precise meaning from the proof, but the statement should also be clear. Maybe introducing some figure, such as Fig. 12, there could improve its readability.

Proposition 2, arguably the central result of the paper, provides a time-delocalised realisation of a unitary tripartite process. The main statements are given in terms of some circuit, so it is not very illuminating to reproduce them in words here.

Finally, in Sec. IV the latter result is applied to the AFBW process, which is known to be unitary (or unitarily extendible) and violate causal inequalities.

The results are definitely interesting, but I'm not sure they meet Nature Communications acceptance criteria. First, the results are rather technical and do not seem accessible to a broad audience. Second, the main result, i.e., the time-delocalised realisation for tripartite unitary process can be considered an incremental result with respect to the bipartite case presented in Ref. 19. In fact, are not able to present a general result for an arbitrary number of parties, in fact they claim it is not known whether this construction is possible. Finally, despite the interest for time-delocalised operations, I have the feeling they do not completely solve the controversy associated with actual experimental realisations of such processes. For instance, the authors write in the Discussion:

"In this work, we showed that, surprisingly, there exist processes that have realisations on time-delocalised subsystems and that violate causal inequalities, a feature that is generally believed to be impossible within standard physics."

I would argue instead that time-delocalised operations are not really considered in "standard physics", so this "surprise" is rather limited. They definitely provide an interesting way of reasoning in terms of these processes, but many problems remain open. To give a further example, the definition of "closed laboratories" discussed in relation to causal inequalities is extremely difficult to justify in all known experimental setups.

In conclusion, I acknowledge the importance of the present results, but I'm not convinced they meet Nature Communications' acceptance criteria. If the authors can convincingly argue about the relevance and general interest of their work, I may reconsider my recommendation.

NCOMMS-22-15531—Existence of processes violating causal inequalities on time-delocalised subsystems

Response to reviewers

Julian Wechs,^{1,2} Cyril Branciard,² and Ognjan Oreshkov¹

¹*QuIC, Ecole Polytechnique de Bruxelles, C.P. 165, Université Libre de Bruxelles, 1050 Brussels, Belgium*

²*Univ. Grenoble Alpes, CNRS, Grenoble INP, Institut Néel, 38000 Grenoble, France*

(Dated: December 20, 2022)

We would like to thank all reviewers for the consideration of our work, and for the constructive and useful feedback. We have made significant changes to our paper in response to Reviewers #2 and #4’s comments (recalling that Reviewer #3 already “strongly recommended publishing [our] paper”, which they considered “a major advance”, and judged “carefully drafted”). In particular, we explained more clearly why the processes we consider can be said to be implementable in standard quantum theory, and further discussed the relevance and broader impact of our work. We have also made various modifications in order to provide clearer explanations regarding the technical aspects pointed out by the reviewers. Below, we provide a detailed response to the individual comments of Reviewers #2 and #4, along with descriptions of the changes we have made. All changes to the text appear in purple in the resubmitted version of our paper.

REVIEWER #2

A. Qualitative evaluation

- *Mathematically, the paper is very sophisticated and solid, except for a few small points that I will ask about below.*

We thank the reviewer for the thorough and detailed reading of our paper, which has led us to improve the manuscript in many respects. We have addressed all of these points (see the technical details section below), and hope that our modifications and replies fully clarify the questions raised.

- *The paper is guaranteed to have impact in the quantum causal modeling field. To have more general impact, the authors should provide a more intuitive interpretation of time-delocalized systems that goes beyond mathematical concepts such as isomorphisms. If one could explain to other physicists what an indefinite causal structure breaking causal inequalities “looks like”, that would certainly be very impressive. It would be particularly spectacular if the authors could propose a table-top quantum-optics experiment that implements those processes, similar to the quantum switch. While such an experiment would necessarily suffer from a similar controversy as the quantum switch experiments, this would mostly be a matter of taste. Several formulations in the paper give the impression that the authors believe that their representation is implementable. However, currently I do not see how to implement their quasi-circuit, while using the agents’ operations only once. I will explain my doubts below.*

We indeed believe that the processes we consider are implementable, in the same sense as the quantum switch, and that their implementation is given by the temporal circuit in Fig. 5 (or Fig. 7 for the Lugano process in particular). In our paper, the focus is not so much on concrete laboratory experiments, but rather on the fundamental question of what it means in principle for a process to be “implementable in standard quantum theory”. This topic is indeed subject to a lot of controversy, as you point out in your comment, which notably concerns questions such as whether and in what sense the relevant operations are used once and only once.

Importantly, we think that the answer to this fundamental question about the standard quantum mechanical realisability of indefinite causal order should not rely on heuristic considerations, or be a matter of taste. Rather, we believe that it should be based on a rigorous theory that relates the standard quantum theoretical description of the alleged implementations (in terms of some temporally ordered quantum circuit) to the description in the abstract process matrix formalism. Such a link is established by the theoretical concept of *time-delocalised subsystem*—the idea is that when the standard, temporal quantum circuits under consideration are described in terms of these alternative (but operationally just as meaningful) systems, they precisely “look like” the situation considered in the process matrix framework (where one instance of each operation is composed with the process matrix in a “circuit

with a cycle”). This argument puts the claim that the quantum switch is implementable in standard quantum theory on a formal basis (as shown in Ref. [1]). The central result of our work is that it extends to unitary extensions of tripartite processes, in particular the Lugano process.

Applying our general tripartite construction to this example, we find the temporal circuit of Fig. 7. There, Alice’s and Bob’s input and output systems are time-delocalised in the same way as in the quantum switch, while Charlie’s input and output systems are time-delocalised in a very different way—intuitively speaking, a time-local instance of U_C is applied once in the beginning of the circuit, and later potentially reversed and reapplied, which notably depends on the output of U_A and U_B . (Importantly, this however does not mean that “Charlie acts multiple times”, see the answer to question 2 below). From our point of view, this circuit provides an answer to the question of what a causal inequality violating experiment can “look like”: From a temporal perspective, i.e., in terms of standard quantum mechanical time-evolution on “time-local” systems, it looks precisely like in Fig. 7. But when one moves to a description in terms of different, time-delocalised systems, it instead looks like the noncausal Lugano process.

It is indeed an interesting question how one could conceive concrete laboratory experiments based on this finding, however this would go beyond the scope of our paper.

In the revised manuscript, we made several changes to highlight the central train of thought of our work, and to provide more intuition behind the calculations. In particular, we rewrote part of the introduction, and added paragraphs with further explanations, for instance at the beginning of Sec. II, as well as Sec. II B, and before Propositions 1 and 2. We also modified the discussion, where we now comment in more detail on the intuition behind the tripartite example (see also the new Appendix F, where we discuss this example in relation to the *closed laboratory* assumption).

To make the calculations of the tripartite proof more intuitively accessible, we furthermore added several figures to illustrate the isomorphisms that define the time-delocalised subsystems (Fig. 15 for the general tripartite case, and Fig. 18 for the specific example of the Lugano process), as well as to illustrate the “circuit rewriting” procedure that takes us from the description in terms of the standard, time-local systems to the description in terms of the time-delocalised systems (Figs. 16 and 17 for the general case, and Figs. 19–22 for the the Lugano process).

B. Technical details

1. Several times, e.g. at the end of page 2, you claim that causal inequalities are device-independent. However, the apparent device-independence is only true assuming some idealizations that may fail in crucial regimes of relativistic quantum information and quantum gravity. Since these are the regimes in which we expect indefinite causal structures to arise in nature, it is crucial to evaluate whether a causal inequality breaks down because of indefinite causal structure, or because the required idealizations fail. This, however, will require theory-specific and physics-specific testing and reasoning. Two crucial examples for these problems with device-independence of causal indefiniteness already exist in the literature. The paper

- C. T. Marco Ho, Fabio Costa, Christina Giarmatzi, Timothy C. Ralph, Violation of a causal inequality in a spacetime with definite causal order, arXiv:1804.05498

shows that the problems of wave-package spreading and localization in quantum field theory can lead to the break down of causal inequalities in an ordinary definite classical background spacetime structure. Furthermore, the recent paper

- Kacper Dębski, Magdalena Zych, Fabio Costa, Andrzej Dragan, Indefinite temporal order without gravity, arXiv:2205.00164

shows that the relativistic phenomenon of entanglement harvesting allows observers to become correlated despite them not being allowed to signal to each other, and despite there being no other common cause than the quantum vacuum which is always present.

It is true that, in order for a causal inequality violation to be a meaningful witness of causal indefiniteness, the assumptions that enter the derivation of causal inequalities must be plausible and compelling, which requires a careful analysis of whether this is the case in the scenario under consideration. This is precisely the topic of Section V in our paper (as well as the further analysis in Appendix F), where we formalise these assumptions for the multipartite case, and analyse our result with respect to these assumptions, leading us to the conclusion that causal inequalities are indeed a meaningful concept to show the absence of a definite causal order between the time-delocalised variables we identified.

In quantum gravity or relativistic quantum information, such an analysis would indeed be subject to the subtleties studied in the papers you pointed out. However, note that here, we have a standard quantum situation, which does not involve any such exotic physics, so these concerns do not directly apply to our case.

In the revised version, we have added a paragraph pointing out that such subtleties have to be taken into account in the situations you mentioned (lines 406–411).

2.

- *Now, let me explain my doubts about the implementability of your quasi-circuits. This involves Figure 5 and Figure 6, and furthermore Figure 7 for the Lugano-process specifically. In Figure 5, U_A and U_B can be implemented via superposition of trajectories, similarly as in table-top experiments of the quantum switch, and are therefore unproblematic, up to the usual problem of fair resource counting. The problem is U_C . We have three gates ω_1 , ω_2 and ω_3 , all of which depend on U_C . Therefore, Figure 5 might use U_C three times. This means that Figure 5 does not really give insights about how to implement the processes physically, while only using each agent operation once. Figure 6 b) and Figure 6 c) use signalling to the past, so also these are not valid implementations. But Figure 6 a) might give some hints about implementations. During the proofs, it does not become too clear what precise U_C -dependence J_{in} , J_{out} and $R(U_C)$ have. Is there a chance to implement a process this way, while only using the agents' operations once? If not, I am out of ideas. How do you think your quasi-circuits help to implement processes? If you do not believe that your schemes can be implemented without using agent operations several times, then what else is it that your quasi-circuits offer?*

The question of whether indefinite causal order is realisable in standard quantum theory, and what a “genuine” implementation of an indefinite causal order process would be, is very controversial in the field. Indeed, from a “valid” implementation, one would require that each input operation of the process is “used once and only once” in some sense, as in the process matrix framework, one instance of each operation appears. Whether this is the case is however far from clear already in the existing implementations of the quantum switch. To justify this, it is usually argued heuristically that each operation occurs precisely once in each of the two coherent “branches” that are superposed, therefore each operation is used once overall, or that one could introduce an additional “flag” or “counter” system that keeps track of how many times each operation has been used.

The time-delocalised systems framework approaches this question from a different angle. Rather than invoking such heuristic considerations, the aim is to establish a formal theory for what it means to implement indefinite causal order processes in standard quantum theory. The idea is that the process matrix picture is obtained when one changes the *systems* with respect to which one describes the implementations under consideration. The new systems thereby are *time-delocalised* (rather than “time-local”, as in the temporal, standard quantum description).

It is important to emphasise that this is not just a mathematical relation between two different mathematical objects: the considered isomorphisms define specific physical subsystems in the experiment, on which the process lives. This is no different from the way we would formally define any physical time-local subsystem on which we perform experiments, if we want to describe where that subsystem is within the larger Hilbert space of, say, quantum field theory. Indeed, we have explained what procedures one would need to perform in practice in order to physically address these time-delocalised systems and experimentally verify that the claimed cyclic circuit from the process matrix picture takes place on these systems. Of course, one could alternatively describe all of these procedures in the language of the “old” systems, but this does not mean that the “new” description is somehow less physical. The exact same thing happens when switching between different choices of factorisation of some physical Hilbert space at a given time.

In other words, the temporal circuit shown in Fig. 7 *is precisely* the cyclic “quasi-circuit” that one considers in the process matrix framework, when one moves to this new perspective. In this cyclic circuit, there is indeed one instance of each operation U_A , U_B and U_C , which are composed with the process matrix. It is just that they do not act on some *time-local* system, but on the time-delocalised input and output systems we identify.

Based on an intuitive reading of the circuit in Fig. 7, one may be led to think that Charlie is involved in several rounds of communication, or “receives and sends out a system several times”. But it is important to note that the Charlie from this intuitive analysis is not the Charlie that acts on the systems C_I and C_O we identify. Instead, the latter Charlie should be understood abstractly as an agent who controls precisely the parameters describing the operation occurring in the time-delocalised slot we have defined for him.

- *If your advantage is in terms of interpretation, then I would like to compare your quasi-circuits to those of Allard-Guerin and Brukner in your reference [2]. Inspired by the concept of time-delocalized systems, they offer a so-called causal reference decomposition of purifiable processes. These are associated with one agent (say, A), who sees a time-evolution of the form $U = U_{future}(U_B, U_C) \cdot U_A \cdot U_{past}(U_B, U_C)$. What is the advantage of your representation over theirs? Also they propose a detailed quasi-circuit for the Lugano/BW/AF-process. Similar to your Figure 7, it has the problem that at least one of the other agents' operations is used several times. In the paper, you claim that the gates are only applied once, because, U_C gets applied, and later on it gets cancelled by U_C^\dagger , and at the end U_C*

is applied again. However, if one tried to implement this circuit gate-after-gate, then one would see the use of three gates that depend on U_C , instead of one or two. I see two crucial advantages in your quasi-circuits over those of Allard-Guerin and Brukner. First of all, up to the issues of resource counting in superpositions of trajectories, your scheme only uses U_A and U_B once. Secondly, your scheme gives an explicit quasi-circuit for all pure three-party processes, up to the $\omega_j(U_C)$ which could in principle be recovered from the constructive proof. Do you see other advantages?

The paper [2] by Guerin and Brukner addresses a question that is different from ours. As you point out, they study causal reference frame representations of the type you describe, which they show to exist for any pure process. The circuit in their Eq. (65) is a particular example of such a representation for the Lugano process, where A is the agent from whose “reference frame perspective” the process “looks local”, and with $U_{\text{past}}(U_B, U_C)$ and $U_{\text{future}}(U_B, U_C)$ having a specific form.

In our work, we are not concerned with such causal reference frame decompositions, but rather with the question of whether there exists a *realisation* of a given processes *on time-delocalised systems*; that is, whether there exists a temporally ordered circuit in which precisely the noncausal process under consideration “happens”—but on some alternative systems, different from the “time-local” ones that one sees in the temporal circuit. For the temporal circuits we consider (Fig. 5 for the general case, and Fig. 7 for the Lugano process), we show that this is indeed the case.

It is quite possible that the circuit in Eq. (65) of Ref. [2] is also a realisation of the Lugano process on some suitably chosen time-delocalised systems—this is, in fact, an interesting open question, and it is not obvious at all whether this is true. To show this, one would need to identify the time-delocalised subsystems (i.e., specify the isomorphisms that define the “new” time-delocalised systems as subsystems of the “old”, time-local ones), and “rewrite” the circuit with respect to these new systems, like we do for our circuits. We currently do not know whether this is possible.

Note that, in Ref. [2], Guerin and Brukner refer to Ref. [1] by Oreshkov, claiming that there exists “a representation in terms of time-delocalised subsystems for [all] multipartite pure processes”. However, this claim is based on a misunderstanding of the concept of time-delocalised subsystem (later clarified in Footnote [58] of Ref. [1]). What Guerin and Brukner show is that, for any multipartite pure process, there exists an abstract decomposition as in our Eq. (C3)/Fig. 12 (for the bipartite case), and Eq. (D16)/Fig. 14 (for the tripartite case). These decompositions indeed generalise straightforwardly to an arbitrary number of parties. But this by itself does not imply that there is a suitable temporal circuit in which these systems can be identified as time-delocalised subsystems of some time-local systems, for which much more work is required.

3. *The general relativity literature already has a name for what you call causal elsewhere: One says X is not causally connected to Y , or X is causally disconnected from Y . You may consider using that term instead.*

Here, we call it “causal elsewhere” following the terminology of *A. S. Eddington, The nature of the physical world (Cambridge University Press, 1928)*. However, in the revised version, we have pointed out that it is also called “not causally connected” or “causally disconnected” (cf. lines 435–436).

4. *It is true that your Free choice-postulate contains the ability to do actions that were not predictable from earlier knowledge. However, your free choice-postulate assumes much more: It assumes no signaling to the (causal) past, and no faster-than-light signaling (i.e. no signaling to the causal “elsewhere”).*

This seems to be primarily a question about terminology. By definition, we say that variables are *freely chosen* if they cannot be correlated with any properties pertaining to their causal past or elsewhere (i.e., the variables that these sets contain, the values of these variables, and their causal order). This intuitive idea is formalised through Eq. (8).

5. *In your Figure 11, one needs to act on internal wires, while the bipartite case only required swaps with open wires. In particular, in the tripartite case it seems that one has to have access to more than just the agents’ inputs and outputs. Does that pose problems for the operational meaning?*

Indeed, in the tripartite case, one first “tests” the structure of the cyclic circuit with respect to the systems P_O , A_I , A_O , B_I , B_O , C_I , C_O , F_I , as well as the additional systems Z and \bar{Z} (i.e., with the systems Y , \bar{Y} , \bar{Q}_1 and \bar{Q}'_2 remaining composed), which is achieved by the modifications shown in Fig. (11). This does however not pose problems for the operational meaning. Namely, with respect to these systems, the circuit consists of operations $U_1 : \mathcal{H}^{P_O A_O B_O} \rightarrow \mathcal{H}^{C_I Z}$, $U_2 : \mathcal{H}^{C_O \bar{Z}} \rightarrow \mathcal{H}^{A_I B_I F_I}$, $\mathbb{1}^{Z \rightarrow \bar{Z}}$, U_A , U_B and U_C (cf. the left-hand side of Eq. (D24)), that is, U_C already appears as an explicit part of the circuit. Therefore, once this structure of the circuit is established as an experimentally verifiable fact, and given that arbitrary U_C can be applied, one can disconnect only the operation U_C , while leaving Z and \bar{Z} composed, and test operationally that the circuit with these systems remaining composed

consists precisely of U , U_A , U_B and U_C .

6. In the beginning of Appendix C 2, about U_1 and U_2 : Intuitively and physically, it makes sense to me that such maps exist. However, I cannot see a mathematical proof that such maps exist/are well-defined, or that the decomposition of Equation (C3) is possible. Do you have a proof? Or does it have to be an extra postulate (it would be a reasonable one)? Furthermore, why are U_1 , U_2 unitary? Is this necessary to make U unitary?

We indeed have a proof that the decomposition of Eq. (C3), with unitary maps U_1 and U_2 , exists for any unitarily extended bipartite process. The proof is given in the paragraph after Equation (C3) (i.e., the paragraph from line 776 to line 788). Briefly summarised, the argument is as follows. We compose the process vector with a SWAP operation (denoted by S_A) in Alice’s “slot”, which results in a unitarily extended one-party process vector for the remaining party Bob. (Figuratively speaking, we “pull Alice’s input wire to the global future, and her output wire to the global past”). A unitarily extended one-party process vector describes a *quantum comb*, which means that it can be decomposed into a circuit. In general, the circuit decomposition of a quantum comb consists in a sequence of *isometric* operations, whose composition is the minimal Stinespring dilation of the comb, and with an ancillary system at the end of the circuit that is traced out. Since the quantum comb we have here is unitary, the ancillary system in the minimal Stinespring dilation is trivial, and the corresponding circuit isometries $\tilde{U}_1 : \mathcal{H}^{A_I P_O} \rightarrow \mathcal{H}^{B_I Z}$ and $\tilde{U}_2 : \mathcal{H}^{B_O Z} \rightarrow \mathcal{H}^{A_O F_I}$ must be unitaries. The decomposition in Eq. (C3) of the original process is then obtained by identifying A'_I with A_O , and A'_O with A_I .

The proof of the corresponding decomposition in the tripartite case (Eq. (D16)) is analogous, with both Alice’s and Bob’s input (resp. output) wires being pulled to the future (resp. past).

In the revised manuscript, we have made some modifications in the paragraph from line 776 to line 788, in order to highlight that it is indeed the proof of Eq. (C3). In the previous Appendix C 1, where we first use the argument with the minimal Stinespring dilation for the operations $\omega_1(U_B)$ and $\omega_2(U_B)$ of the temporal circuit, we have elaborated on the explanation of why they can be taken to be unitary, and we refer to it in Appendix C 2.

7. In Appendix C 2: Why is $|U_{\mathcal{G}}(S_A, \cdot)\rangle\rangle$ a valid process vector?

We start from a valid bipartite unitarily extended process vector $|U\rangle\rangle$, which is known to yield a unitary transformation from $\mathcal{H}^{A'_I B'_I P_O}$ to $\mathcal{H}^{A'_O B'_O F_I}$ when composed with arbitrary unitary operations $U_A : \mathcal{H}^{A_I A'_I} \rightarrow \mathcal{H}^{A_O A'_O}$ and $U_B : \mathcal{H}^{B_I B'_I} \rightarrow \mathcal{H}^{B_O B'_O}$ performed by Alice and Bob. To obtain $|U_{\mathcal{G}}(S_A, \cdot)\rangle\rangle$, we now “fix” Alice’s operation to be the particular unitary S_A , i.e., we consider the composition $|U_{\mathcal{G}}(S_A, \cdot)\rangle\rangle = |U\rangle\rangle * |S_A\rangle\rangle$. This process vector $|U_{\mathcal{G}}(S_A, \cdot)\rangle\rangle$ with a “fixed” operation for Alice still yields a unitary transformation from $\mathcal{H}^{A'_I B'_I P_O}$ to $\mathcal{H}^{A'_O B'_O F_I}$ when composed with an arbitrary unitary operation $U_B : \mathcal{H}^{B_I B'_I} \rightarrow \mathcal{H}^{B_O B'_O}$ for Bob. Therefore, $|U_{\mathcal{G}}(S_A, \cdot)\rangle\rangle$ can be seen as a valid unitarily extended one-party process (for Bob), where the output systems of the “global past” party are now taken to be $A'_I P_O$, and where the input systems of the “global future” party are now taken to be $A'_O F_I$. Note that we use the same argument for $|U_{\mathcal{G}}(\cdot, U_B)\rangle\rangle$ in the previous subsection C 1 (cf. lines 751 to 753).

8. Can you explain in more detail why Equation (D3) is true? It looks like an expansion of the brackets leads to cross-terms like $|\nu_1^{A \prec B}\rangle\rangle * |\nu_2^{A \prec B}\rangle\rangle$.

We have made some more extensive changes to Appendix D 1 in order to explain the ideas there in a clearer way (see also our answer to question 9 below).

The relevant equation which defines the “enlarged” process is now Eq. (D8). Indeed, when inserting the “enlarged” operations (Eqs. (D2)-(D7)) and expanding the terms on the right-hand side of Eq. (D8), one would find cross terms between the original and the “complementary” operations (in the new version denoted by $\nu_{1,\text{comp}}^{A \prec B}$ etc.), such as $\dots |\nu_1^{A \prec B}\rangle\rangle * |\nu_{2,\text{comp}}^{A \prec B}\rangle\rangle \dots$. However, the corresponding link products evaluate to zero. This is because $\nu_1^{A \prec B}$ and $\nu_{2,\text{comp}}^{A \prec B}$ are defined so as to act on orthogonal subspaces of the Hilbert space \mathcal{H}^{E_1} over which they are composed. (The output Hilbert space of $\nu_1^{A \prec B}$ is $\mathcal{H}^{A_I} \otimes \mathcal{H}^{\lambda_1}$, and the input Hilbert space of $\nu_{2,\text{comp}}^{A \prec B}$ is $\mathcal{H}^{A_O} \otimes \mathcal{H}^{\rho_1}$, where \mathcal{H}^{λ_1} and \mathcal{H}^{ρ_1} are complementary orthogonal subspaces of \mathcal{H}^{E_1}). Similarly for the other “cross terms” between the original and “complementary” operations. In the new version of the manuscript, we have added a clarifying comment at the relevant place, where we describe how to recover $|U\rangle\rangle$ from $|\tilde{U}\rangle\rangle$ (line 877 to 882).

9. It is not obvious to me how Equation (D3) solves the problem of different sized ancillas. Is this still a unitary process? How exactly does $|U\rangle\rangle$ get enlarged to act on the other ancillas? Why does U^{comp} help? Why is it necessary?

In the new version of the manuscript, we have changed the notations and explanation, in a way that now appears

more instructive to us. We hope that these modifications clarify these questions. In particular, we spell out explicitly all Hilbert spaces that the operations act upon, so that it is easier to keep track of the various spaces and dimensions involved.

The issue is that in the original process $|U\rangle\rangle$, we have two coherent “branches” that are superposed, corresponding to the orders $A \prec B$ and $B \prec A$, which involve ancillas of different dimensions (λ_1 and λ_2 for the “ $A \prec B$ branch” and ρ_1 and ρ_2 for the “ $B \prec A$ branch”, respectively, with $d_{\rho_1} = d_{\rho_2} =: d_\rho$ and $d_{\lambda_1} = d_{\lambda_2} =: d_\lambda$, but $d_\rho \neq d_\lambda$ in general). We embed the original process into a larger process, where the dimensions of the ancillary systems are the same. (We could alternatively also do without this, but then we would need to consider operations that act unitarily not on the full input and output Hilbert spaces, but on different-dimensional subspaces depending on the respective coherent “branch”. It would be rather tedious to work with such operations, therefore we introduce the enlarged process here.) We take the ancillary spaces in this “enlarged” process to be the direct sums $\mathcal{H}^{E_1} := \mathcal{H}^{\lambda_1} \oplus \mathcal{H}^{\rho_1}$ and $\mathcal{H}^{E_2} := \mathcal{H}^{\lambda_2} \oplus \mathcal{H}^{\rho_2}$. To define it, to each operation in the original process, we consider a “complementary” operation, whose input and output spaces are precisely the subspaces orthogonal to those on which the original operation acts (for instance, we have $\nu_1^{A \prec B} : \mathcal{H}^{P_o^\ell} \rightarrow \mathcal{H}^{A_I} \otimes \mathcal{H}^{\lambda_1}$, and $\nu_{1,\text{comp.}}^{A \prec B} : \mathcal{H}^{P_o^r} \rightarrow \mathcal{H}^{A_I} \otimes \mathcal{H}^{\rho_1}$, with $\mathcal{H}^{P_o^\ell} \oplus \mathcal{H}^{P_o^r} = \mathcal{H}^{P_o}$, and $\mathcal{H}^{A_I} \otimes \mathcal{H}^{\lambda_1} \oplus \mathcal{H}^{A_I} \otimes \mathcal{H}^{\rho_1} = \mathcal{H}^{A_I} \otimes \mathcal{H}^{E_1}$). To then embed the original and complementary operations into a larger process, we consider the additional systems p and f , and define the “enlarged” operations $\tilde{\nu}_1^{A \prec B}$ etc. (Eq. (D2) - (D7)), which we combine to the enlarged process of Eq. (D8). This is a valid process vector, because it is also of the form as in Eq. (D1), which is also sufficient for a vector to define a valid unitarily extended bipartite process (a fact that we now highlight in the new version when introducing the decomposition in Eq. (D1), cf. line 829 and 830), and it is such that the ancillary spaces in both coherent “branches” are the same (namely, E_1 and E_2).

10. In Equation (D4), I cannot see that ω_1 is unitary.

$\omega_1 : \mathcal{H}^{P_{Op}} \rightarrow \mathcal{H}^{T_1 E_1 Q_1}$ is defined (with the new notations, now in Eq. (D9)) as

$$\omega_1 = [(\mathbb{1}^{E_1} \otimes \mathbb{1}^{A_I \rightarrow T_1}) \cdot \tilde{\nu}_1^{A \prec B}] \otimes |0\rangle^{Q_1} + [(\mathbb{1}^{E_1} \otimes \mathbb{1}^{B_I \rightarrow T_1}) \cdot \tilde{\nu}_1^{B \prec A}] \otimes |1\rangle^{Q_1},$$

with $\tilde{\nu}_1^{A \prec B}$ defined in the new Eq.(D2), and $\tilde{\nu}_1^{B \prec A}$ defined in the new Eq.(D5). Let us show by explicit calculation that $\omega_1^\dagger \cdot \omega_1 = \mathbb{1}^{P_{Op}}$, and that $\omega_1 \cdot \omega_1^\dagger = \mathbb{1}^{T_1 E_1 Q_1}$.

$$\begin{aligned} \omega_1^\dagger \cdot \omega_1 &= \{ [(\tilde{\nu}_1^{A \prec B})^\dagger \cdot (\mathbb{1}^{E_1} \otimes \mathbb{1}^{T_1 \rightarrow A_I}) \cdot] \otimes \langle 0|^{Q_1} + [(\tilde{\nu}_1^{B \prec A})^\dagger \cdot (\mathbb{1}^{E_1} \otimes \mathbb{1}^{T_1 \rightarrow B_I}) \cdot] \otimes \langle 1|^{Q_1} \} \\ &\quad \cdot \{ [(\mathbb{1}^{E_1} \otimes \mathbb{1}^{A_I \rightarrow T_1}) \cdot \tilde{\nu}_1^{A \prec B}] \otimes |0\rangle^{Q_1} + [(\mathbb{1}^{E_1} \otimes \mathbb{1}^{B_I \rightarrow T_1}) \cdot \tilde{\nu}_1^{B \prec A}] \otimes |1\rangle^{Q_1} \} \\ &= (\tilde{\nu}_1^{A \prec B})^\dagger \cdot \tilde{\nu}_1^{A \prec B} + (\tilde{\nu}_1^{B \prec A})^\dagger \cdot \tilde{\nu}_1^{B \prec A} \\ &= [(\nu_1^{A \prec B})^\dagger \otimes |0\rangle^p + (\nu_{1,\text{comp.}}^{A \prec B})^\dagger \otimes |1\rangle^p] \cdot [\nu_1^{A \prec B} \otimes \langle 0|^p + \nu_{1,\text{comp.}}^{A \prec B} \otimes \langle 1|^p] \\ &\quad + [(\nu_1^{B \prec A})^\dagger \otimes |0\rangle^p + (\nu_{1,\text{comp.}}^{B \prec A})^\dagger \otimes |1\rangle^p] \cdot [\nu_1^{B \prec A} \otimes \langle 0|^p + \nu_{1,\text{comp.}}^{B \prec A} \otimes \langle 1|^p]. \end{aligned}$$

Noting that $(\nu_1^{A \prec B})^\dagger \cdot \nu_{1,\text{comp.}}^{A \prec B} = 0$ (since the output space $\mathcal{H}^{A_I} \otimes \mathcal{H}^{\rho_1}$ of $\nu_{1,\text{comp.}}^{A \prec B}$ is orthogonal to the input space $\mathcal{H}^{A_I} \otimes \mathcal{H}^{\lambda_1}$ of $(\nu_1^{A \prec B})^\dagger$), and similarly for the other “cross terms” between the original and complementary operations $(\nu_{1,\text{comp.}}^{A \prec B})^\dagger \cdot \nu_1^{A \prec B}$, $(\nu_1^{B \prec A})^\dagger \cdot \nu_{1,\text{comp.}}^{B \prec A}$ and $(\nu_{1,\text{comp.}}^{B \prec A})^\dagger \cdot \nu_1^{B \prec A}$, by evaluating the matrix multiplications, we obtain

$$\begin{aligned} \omega_1^\dagger \cdot \omega_1 &= (\nu_1^{A \prec B})^\dagger \cdot \nu_1^{A \prec B} \otimes |0\rangle\langle 0|^p + (\nu_{1,\text{comp.}}^{A \prec B})^\dagger \cdot \nu_{1,\text{comp.}}^{A \prec B} \otimes |1\rangle\langle 1|^p + \\ &\quad (\nu_1^{B \prec A})^\dagger \cdot \nu_1^{B \prec A} \otimes |0\rangle\langle 0|^p + (\nu_{1,\text{comp.}}^{B \prec A})^\dagger \cdot \nu_{1,\text{comp.}}^{B \prec A} \otimes |1\rangle\langle 1|^p \\ &= \mathbb{1}^{P_o^\ell} \otimes |0\rangle\langle 0|^p + \mathbb{1}^{P_o^r} \otimes |1\rangle\langle 1|^p + \mathbb{1}^{P_o^r} \otimes |0\rangle\langle 0|^p + \mathbb{1}^{P_o^\ell} \otimes |1\rangle\langle 1|^p = \mathbb{1}^{P_{Op}}. \end{aligned} \tag{1}$$

For $\omega_1 \cdot \omega_1^\dagger$, we obtain

$$\begin{aligned} \omega_1 \cdot \omega_1^\dagger &= \{ [(\mathbb{1}^{E_1} \otimes \mathbb{1}^{A_I \rightarrow T_1}) \cdot \tilde{\nu}_1^{A \prec B}] \otimes |0\rangle^{Q_1} + [(\mathbb{1}^{E_1} \otimes \mathbb{1}^{B_I \rightarrow T_1}) \cdot \tilde{\nu}_1^{B \prec A}] \otimes |1\rangle^{Q_1} \} \\ &\quad \cdot \{ [(\tilde{\nu}_1^{A \prec B})^\dagger \cdot (\mathbb{1}^{E_1} \otimes \mathbb{1}^{T_1 \rightarrow A_I}) \cdot] \otimes \langle 0|^{Q_1} + [(\tilde{\nu}_1^{B \prec A})^\dagger \cdot (\mathbb{1}^{E_1} \otimes \mathbb{1}^{T_1 \rightarrow B_I}) \cdot] \otimes \langle 1|^{Q_1} \} \end{aligned}$$

Noting that $\tilde{\nu}_1^{A \prec B} \cdot (\tilde{\nu}_1^{B \prec A})^\dagger = 0$, since the output space $\mathcal{H}^{\tilde{P}_o^r} = \mathcal{H}^{P_o^r} \otimes \text{span}\{|0\rangle^p\} \oplus \mathcal{H}^{P_o^\ell} \otimes \text{span}\{|1\rangle^p\}$ of $(\tilde{\nu}_1^{B \prec A})^\dagger$ is orthogonal to the input space $\mathcal{H}^{\tilde{P}_o^\ell} = \mathcal{H}^{P_o^\ell} \otimes \text{span}\{|0\rangle^p\} \oplus \mathcal{H}^{P_o^r} \otimes \text{span}\{|1\rangle^p\}$ of $\tilde{\nu}_1^{A \prec B}$, and similarly for $\tilde{\nu}_1^{B \prec A} \cdot (\tilde{\nu}_1^{A \prec B})^\dagger$,

we further obtain

$$\begin{aligned}
\omega_1 \cdot \omega_1^\dagger &= [(\mathbb{1}^{E_1} \otimes \mathbb{1}^{A_I \rightarrow T_1}) \cdot \tilde{\nu}_1^{A \leftarrow B} \cdot (\tilde{\nu}_1^{A \leftarrow B})^\dagger \cdot (\mathbb{1}^{E_1} \otimes \mathbb{1}^{T_1 \rightarrow A_I}) \cdot] \otimes |0\rangle\langle 0|^{Q_1} \\
&\quad + [(\mathbb{1}^{E_1} \otimes \mathbb{1}^{B_I \rightarrow T_1}) \cdot \tilde{\nu}_1^{B \leftarrow A} \cdot (\tilde{\nu}_1^{B \leftarrow A})^\dagger \cdot (\mathbb{1}^{E_1} \otimes \mathbb{1}^{T_1 \rightarrow B_I}) \cdot] \otimes |1\rangle\langle 1|^{Q_1} \\
&= [(\mathbb{1}^{E_1} \otimes \mathbb{1}^{A_I \rightarrow T_1}) \cdot \mathbb{1}^{E_1 A_I} \cdot (\mathbb{1}^{E_1} \otimes \mathbb{1}^{T_1 \rightarrow A_I}) \cdot] \otimes |0\rangle\langle 0|^{Q_1} \\
&\quad + [(\mathbb{1}^{E_1} \otimes \mathbb{1}^{B_I \rightarrow T_1}) \cdot \mathbb{1}^{E_1 B_I} \cdot (\mathbb{1}^{E_1} \otimes \mathbb{1}^{T_1 \rightarrow B_I}) \cdot] \otimes |1\rangle\langle 1|^{Q_1} \\
&= \mathbb{1}^{T_1 E_1} \otimes |0\rangle\langle 0|^{Q_1} + \mathbb{1}^{T_1 E_1} \otimes |1\rangle\langle 1|^{Q_1} = \mathbb{1}^{T_1 E_1 Q_1}.
\end{aligned} \tag{2}$$

Therefore, ω_1 is unitary.

11. *Can you prove that Figure 7 indeed gives the Lugano/Baumeler-Wolf/Araujo-Feix process?*

The proof that Fig. 7 is indeed a realisation of the Lugano process on time-delocalised subsystems consists in applying the general tripartite “circuit rewriting” procedure (represented graphically in Fig. 6 and detailed mathematically in Appendix D 3) to that particular temporal circuit (modulo some simplifications in the way the ancillary systems in the circuit are treated, see Sec. IV), and with particular isomorphisms J_{in} and J_{out} . In the manuscript, in the interest of brevity and conciseness, we have opted against working this out explicitly, since it would essentially just be a repetition of the calculations in Appendix D 3, with a particular form for ω_1 , ω_2° , ω_2^\bullet , ω_3 , J_{in} and J_{out} . However, note that we specify the form of these operations in Appendix D 4a.

In the new version of the manuscript, we have added a comment at the end of this appendix (cf. lines 982 to 989) to outline how the calculation would be done, and to highlight more clearly that the operations given there are the specific operations that appear in applying the general tripartite proof to this particular example. We also added several figures illustrating the proof for this example (Figs. 18–22; see in particular the caption of the latter figure).

12. *Appendix E seems overcomplicated. Your free choice-requirement also includes a condition that there is no signalling to the causal past and the causal “elsewhere”. In other words, it says there can only be signalling to the causal future. Equation (8) just seems like a rephrasing of no-signalling-outside-the-causal-future, and so does Equation (E 3).*

See the reply to question 4.

REVIEWER #4

- *In conclusion, I acknowledge the importance of the present results, but I’m not convinced they meet Nature Communications’ acceptance criteria. If the authors can convincingly argue about the relevance and general interest of their work, I may reconsider my recommendation.*

We thank reviewer #4 for the consideration of our work, and we are pleased that the reviewer acknowledges the importance of our results. To make a case for the acceptance of our paper, let us summarise again the main conceptual contributions of our paper, and explain why we believe they represent breakthrough results for the field of quantum causality that are of interest for a broad audience.

Our work is concerned with the question of which causally indefinite processes are physically meaningful or implementable in practice—and, more particularly, which of these processes can be realised in standard quantum theory, without resorting to exotic physical regimes. This question is crucial both from a foundational perspective and for potential applications of indefinite causal order, and therefore, it is among the most important and hotly debated ones in the field. There have been numerous laboratory experiments claiming to implement processes with indefinite causal order, notably the “quantum switch”, which is based on coherent control of the order in which the operations are applied. The interpretation of these experiments as “valid” implementations of indefinite causal order has however been disputed.

We believe that, to properly address the question of standard quantum realisability of indefinite causal order, one should resort to a rigorous theory of what it means to “implement an indefinite causal order process in standard quantum theory” (rather than relying on the somewhat heuristic arguments that are usually invoked). That is, one needs a theory which relates the temporal, standard quantum theoretical description of the implementations under consideration to their description in the abstract process matrix framework, in which indefinite causal order is formally described. Such a theory was developed in Ref. [1], based on the concept of *time-delocalised subsystems*. What we found in our work is that this framework of time-delocalised subsystems allows for the realisation of much more radical

situations than those based on coherent control of orders, showing that the latter paradigm is not sufficient to capture all possibilities. Notably, certain processes which violate *causal inequalities* have realisations on time-delocalised subsystems. Whether such processes admit any kind of physical realisation has been one of the most central questions in the field of indefinite causal order since its beginning. In our paper, we show that by properly defining what it means for an indefinite causal order process to “have a realisation in standard quantum theory”, one comes to the conclusion that such process matrices actually can be realised. We analyse this finding with respect to the *free choice* and *closed laboratory* assumptions which underlie causal inequalities, and find that these assumptions are not violated in any manifest way. Therefore, this violation of a causal inequality can be seen as a compelling, device-independent demonstration of the nonexistence of a possibly dynamical and random causal order between the variables.

The resolution of this central question opens up all kinds of new avenues for research, both with regard to the fundamental implications (e.g. in connection to quantum reference frames), and with regard to possible applications (e.g. in device-independent quantum cryptography). We therefore think that it is a result of high impact and wide interest.

We also thank Reviewer #4 for the additional comments and questions, the consideration of which has allowed us to further improve our paper. Please find our replies below.

- *First, the results are rather technical and do not seem accessible to a broad audience.*

It is true that the tripartite proof involves various technicalities, but we have intentionally structured the paper in such a way that the main text, which only presents the results, can be read and understood independently of the appendices, in which we provide the technical details of the proofs. In the new version, we have made further efforts to make the main text more accessible by explaining more clearly the central trains of thought, by highlighting the conceptual significance of our results and by providing more detailed explanations about the intuitions behind the calculations. For that purpose, we rewrote the introduction and discussion, and added explanatory paragraphs in several places (notably at the beginning of Sec. II, at the beginning of Sec. II B, and before Propositions 1 and 2).

- *Second, the main result, i.e., the time-delocalised realisation for tripartite unitary process can be consider an incremental result with respect to the bipartite case presented in Ref. [1]. In fact, are not able to present a general result for an arbitrary number of parties, in fact they claim it is not known whether this construction is possible.*

It is true that we do not have a proof for general N -partite unitarily extendible processes, or any other class beyond unitary extendible tripartite processes, which would be highly desirable. However, the step from two to three parties is not only a small technical increment, but has far-reaching conceptual consequences. The qualitative difference between bipartite and tripartite unitarily extended processes is that the latter can violate causal inequalities, while the former cannot. (It has been shown that all bipartite unitarily extended processes are variations of the quantum switch [3, 4]. This means that they fall into the class of processes that can be implemented via coherent control of the order of operations, which can only generate causal correlations [5, 6].) Therefore, our paper shows that there are processes which have realisations in a well-defined sense, but which do not fall into the quantum controlled paradigm, and require qualitatively new ways of “time-delocalising” quantum information.

- *Finally, despite the interest for time-delocalised operations, I have the feeling they do not completely solve the controversy associated with actual experimental realisations of such processes. For instance, the authors write in the Discussion: “In this work, we showed that, surprisingly, there exist processes that have realisations on time-delocalised subsystems and that violate causal inequalities, a feature that is generally believed to be impossible within standard physics.” I would argue instead that time-delocalised operations are not really considered in “standard physics”, so this “surprise” is rather limited. They definitely provide an interesting way of reasoning in terms of these processes, but many problems remain open.*

By the above statement, we mean that the processes we consider can be realised with the tools of standard physics, without needing to resort to unknown physical regimes or exotic, “nonstandard” physical concepts (such as, for instance, at the interface of quantum theory and gravity).

It is true that time-delocalised subsystems are not usually considered in quantum foundations or quantum information—they are a new concept, first introduced in Ref. [1], and then further developed in our present paper. But this new concept is solidly anchored within the usual quantum theoretical formalism—it simply arises from combining two notions of standard quantum theory that are well established. On the one hand, we use the

fact that we can think of quantum circuit fragments as implementing operations from all of their incoming systems (which are generally associated with different times) to all of their outgoing systems (generally also associated with different times). On the other hand, we use the fact that quantum subsystem decompositions are defined in terms of tensor product structures on the relevant Hilbert spaces. Combining these two notions, it turns out that that a given quantum circuit can be described with respect to an alternative choice of systems, which is operationally equally meaningful.

We thus do not leave the framework of conventional quantum theory—rather, the idea is that, *within* the established quantum theory, we can have more general types of systems and operations than those usually considered, which provides a more general way in which we can study realisations of causally indefinite processes.

- *To give a further example, the definition of “closed laboratories” discussed in relation to causal inequalities is extremely difficult to justify in all known experimental setups.*

The closed laboratory assumption is one of the two constraints that enter the derivation of causal inequalities. Its precise technical formulation was first developed in Ref. [7], for the case of two parties. Here, we formulate these assumptions in the multipartite case. Our formulation involves a relaxed form of the closed laboratory assumption, and therefore a strengthening of the original derivation in Ref. [7]. In the example we study, we then find that the closed laboratory assumption is not violated in any manifest way—rather, based on the observable causal relations between the time-delocalised variables, it is plausible to believe that the closed laboratory and free choice assumptions hold (see the discussion in Sec. V). This means that the causal inequality violation we find is indeed compelling and meaningful as a device-independent concept to demonstrate the absence of a causal order.

In the revised version, we have added a paragraph at the end of Sec. V (lines 526–532), as well as a paragraph in the discussion (lines 558–578) and a more detailed Appendix F. There, we further comment on the closed laboratory assumption, and why it is not apparent that it is violated in our setting. Namely, based on an intuitive analysis of the temporal circuit in our example (Fig. 7), one could come to the conclusion that several rounds of information exchange take place, and therefore the closed laboratory assumptions may appear to be violated. However, in this intuitive analysis, one implicitly associates certain systems and operations with the three parties, which turn out not to be the time-delocalised systems and operations that define the parties that act on the time-delocalised subsystems we identify.

- *Proposition 1 shows the existence of a specific construction for a bipartite unitary process in terms of a localised operation for Alice and a delocalised operation for Bob (point 1). The second point, similarly, concerns some time-delocalised circuit implementation. I must say that it is rather confusing in its formulation. Of course, one can recover its precise meaning from the proof, but the statement should also be clear. Maybe introducing some figure, such as Fig. 12, there could improve its readability.*

The figures that illustrate the result in Proposition 1 are Fig. 3 (which shows the temporal circuit that realises unitarily extensions of bipartite processes) and Fig. 4 (which shows how to describe this circuit in terms of the new, time-delocalised subsystems). Similarly, the figures that illustrate the result in Proposition 2 are Fig. 5 and Fig. 6. We now added an explanatory paragraph before each of the respective propositions (lines 251–255, and lines 299–303, respectively), which gives an intuitive account of the results that are formalised by the respective propositions. We also refer directly to Fig. 4 and Fig. 6 in Propositions 1 and 2, in order to highlight more clearly that they pertain to these respective propositions.

Note also that we added two further figures that provide an intuition for how we obtain the isomorphisms that define the relevant time-delocalised subsystems in our tripartite proofs (Fig. 15 for the general tripartite case, and Fig. 18 for the particular case of the Lugano process).

-
- [1] O. Oreshkov, Quantum **3**, 206 (2019), arXiv:1801.07594 [quant-ph].
 - [2] P. A. Guérin and Č. Brukner, New J. Phys. **20**, 103031 (2018), arXiv:1805.12429 [quant-ph].
 - [3] J. Barrett, R. Lorenz, and O. Oreshkov, Nat. Commun. **12**, 885 (2021), arXiv:2002.12157 [quant-ph].
 - [4] W. Yokojima, M. T. Quintino, A. Soeda, and M. Murao, Quantum **5**, 441 (2021), arXiv:2003.05682 [quant-ph].
 - [5] J. Wechs, H. Dourdent, A. A. Abbott, and C. Branciard, PRX Quantum **2**, 030335 (2021), arXiv:2101.08796 [quant-ph].

- [6] T. Purves and A. J. Short, *Phys. Rev. Lett.* **127**, 110402 (2021), arXiv:2101.09107 [quant-ph].
- [7] O. Oreshkov, F. Costa, and Č. Brukner, *Nat. Commun.* **3**, 1092 (2012), arXiv:1105.4464 [quant-ph].

REVIEWERS' COMMENTS

Reviewer #2 (Remarks to the Author):

The authors have taken all my mathematical questions into account and adapted the manuscript.

Another reviewer complained that this work is only an incremental improvement over the bipartite case. I disagree with this claim. The bipartite case has very special structure, and already the tripartite case is notoriously difficult. This also shows in the present paper, which is mathematically very impressive.

Thanks to the authors' changes to the manuscript, I think I now understand better the concept of time-delocalized systems: A factorization of the full composite Hilbert space of all ingoing/outgoing systems into constituent systems. The isomorphisms $J_{\{in\}}$ and $J_{\{out\}}$ just put the terms "factorization" and "delocalization" on a solid mathematical foundation. It makes sense that they can depend on the agents' operations, because one agent might control how another agent gets delocalized. One example for this is the control qubit agent in the quantum switch.

Therefore, I agree with the authors that time-delocalized systems are a way how to imagine indefinite causal structures: The isomorphisms describe the time-delocalization itself, and the process after the isomorphisms gives the indefinite causal structure on the now localized systems (that were originally time-delocalized).

Furthermore, the authors added several figures and clarifications for the crucial example of the Lugano/BW-process. This is very helpful. Very much to my surprise, the isomorphisms $J_{\{in\}}$ and $J_{\{out\}}$ that translate between time-local and time-delocalized systems do not seem to depend on any of the agents operations! I think that alone is a crucial insight that was not obvious in the previous version.

Since the authors worked on all of my concerns, I do not have further wishes for changes. So from my side, the manuscript is ready for publication.

However, I am very conflicted about whether Nature Communications is the right journal for publication, because of its interdisciplinary audience. On one hand, the visualization of exotic strong non-causal processes via time-delocalized systems is very nice, and very intuitive once properly understood. On the other hand, it needs a significant mathematical and quantum causal background to be appreciated. Furthermore, the insights do not seem to offer much to experimentalists: The main circuits such as in Figure 7 and 21 would probably still have to be implemented experimentally by using U_C two times, and U_C^\dagger one time. The fact that the resulting statistics is equivalent to a causally indefinite process that only uses U_C once will only offer little comfort for experimentalists.

Reviewer #4 (Remarks to the Author):

The authors addressed all my comments in a satisfactory way. I have no further objections to publication.